# Expressiveness of Multi-Neuron Convex Relaxations in Neural Network Certification

**Yuhao Mao**[†*], **Yani Zhang**[‡*] **& Martin Vechev**[†]
[†]Department of Computer Science
[‡]Department of Information Technology and Electrical Engineering
ETH Zürich, Switzerland
{yuhao.mao,martin.vechev}@inf.ethz.ch, yanizhang@mins.ee.ethz.ch

## Abstract

Neural network certification methods heavily rely on convex relaxations to provide robustness guarantees. However, these relaxations are often imprecise: even the most accurate single-neuron relaxation is incomplete for general ReLU networks, a limitation known as the *single-neuron convex barrier*. While multi-neuron relaxations have been heuristically applied to address this issue, two central questions arise: (i) whether they overcome the convex barrier, and if not, (ii) whether they offer theoretical capabilities beyond those of single-neuron relaxations. In this work, we present the first rigorous analysis of the expressiveness of multi-neuron relaxations. Perhaps surprisingly, we show that they are inherently incomplete, even when allocated sufficient resources to capture finitely many neurons and layers optimally. This result extends the single-neuron barrier to a *universal convex barrier* for neural network certification. On the positive side, we show that completeness can be achieved by either (i) augmenting the network with a polynomial number of carefully designed ReLU neurons or (ii) partitioning the input domain into convex sub-polytopes, thereby distinguishing multi-neuron relaxations from single-neuron ones which are unable to realize the former and have worse partition complexity for the latter. Our findings establish a foundation for multi-neuron relaxations and point to new directions for certified robustness, including training methods tailored to multi-neuron relaxations and verification methods with multi-neuron relaxations as the main subroutine.

## 1 Introduction

Neural networks are vulnerable to adversarial attacks (Szegedy et al., 2014), where a small perturbation to the input can lead to misclassification. Adversarial robustness, which measures the robustness of a model with respect to adversarial perturbations, has received much research attention in recent years. However, computing the exact adversarial robustness of a general neural network is coNP-hard (Katz et al., 2017), while adversarial attacks (Carlini & Wagner, 2017; Tramèr et al., 2020) that try to find an adversarial perturbation can only provide a heuristic upper bound on the robustness of the model. To tackle this issue, neural network certification has been proposed to provide robustness guarantees. In the context of robustness certification, the task boils down to providing a numerical bound on the output of a neural network for all possible inputs within a given set. A central property of certification is *completeness*, which requires the method to provide exact bounds for all cases.

Certification methods based on convex relaxations can provide efficient certification by computing an over-approximation of the feasible output set of a given network, with certain trade-off on the precision (Wong & Kolter, 2018; Singh et al., 2018; Weng et al., 2018; Gehr et al., 2018; Xu et al., 2020). They can also be incorporated in the training process to deliver models that are easy to certify (Shi et al., 2021; Müller et al., 2023; Mao et al., 2023; 2025; Palma et al., 2024; Balauca et al., 2025). Due to the central role of convex relaxations in the context of certified robustness, it is crucial to understand their theoretical properties.

---

[*]Equal contribution

**The Single-Neuron Convex Barrier**   Single-neuron relaxations are widely studied due to their popularity and simplicity. However, the single-neuron convex barrier (Salman et al., 2019; Palma et al., 2021) prevents single-neuron convex relaxations from providing exact bounds for general ReLU networks. Baader et al. (2024) further show that even the most precise single-neuron relaxation, namely Triangle (Wong & Kolter, 2018), cannot exactly bound any ReLU network encoding the "max" function in $\mathbb{R}^2$. To overcome this limitation, multi-neuron relaxations have been proposed (Singh et al., 2018; Müller et al., 2022; Zhang et al., 2022), achieving higher empirical precision. Yet, their theoretical properties remain largely unexplored. In particular, it is unclear whether multi-neuron relaxations are able to provably bypass the convex barrier and provide complete certification for general ReLU networks, if given sufficient resources. A key challenge is that, unlike the single-neuron setting—where proving a barrier only requires exhibiting a concrete network for which the most precise single-neuron relaxation fails—a multi-neuron relaxation can always be made more precise by allocating more resources, thus this question cannot be answered via empirical studies. Moreover, solving multi-neuron relaxations is significantly more computationally expensive, making empirical exploration of their limits difficult.

**This Work: Quantifying the Expressiveness and Completeness of Multi-Neuron Relaxations** In this work, we formalize the notion of multi-neuron relaxations and rigorously investigate their expressiveness. We address two central questions: (i) whether they overcome the single-neuron convex barrier, and if not, (ii) whether they offer fundamental advantages over single-neuron relaxations.

**Key Contributions**

- We prove that multi-neuron relaxations are inherently incomplete for general ReLU networks, even provided with sufficient resources to capture all neurons in each individual layer optimally (§3). This incompleteness result is extended to relaxations involving finitely many layers and networks with non-polynomial activations, e.g., tanh, establishing a universal convex barrier for neural network certification with convex relaxations (§4).
- We prove that with equivalence-preserving network transformations, a layerwise multi-neuron relaxation can be a complete verifier, which is impossible for any single-neuron relaxation. This shows that the expressivity of general ReLU networks is preserved under multi-neuron relaxations: every continuous piecewise linear function can be encoded by a network that is exactly bounded by some layerwise multi-neuron relaxation (§5.1). This stands in sharp contrast to the impossibility result established for single-neuron relaxations (Baader et al., 2024): in a case study, we demonstrate that a simple network implementing the "max" function in $[0, 1]^d$ can be exactly bounded by a dimension-independent multi-neuron relaxation far weaker than required by the general theorem.
- We analyze the properties of multi-neuron relaxations under convex polytope partitioning and show that their partition complexity required to achieve complete certification is strictly lower than that of single-neuron relaxations (§5.2).
- We discuss the practical implications of the above theorems, including training strategies tailored to multi-neuron relaxations and verification methods with multi-neuron relaxations as the main subroutine (§6).

Aside from the prior works mentioned, an extended discussion of related work can be found in §A.

## 2   BACKGROUND

### 2.1   CONVEX RELAXATIONS FOR CERTIFICATION

Given a function $f : \mathbb{R}^d \to \mathbb{R}^{d'}$ and a compact domain $X \subseteq \mathbb{R}^d$, we denote the graph of the function $\{(x, f(x) : x \in X\}$ by $f[X]$. The certification task boils down to computing the upper and lower bounds of $f(X) := \{f(x) | x \in X\}$, in order to verify that these bounds meet certain requirements, e.g., adversarial robustness. To this end, convex relaxations approximate $f[X]$ by conditioned convex polytopes $S \subseteq \mathbb{R}^{d+d'}$ satisfying $S \supseteq f[X]$, where the condition depends on the concrete relaxation method. We then take the upper

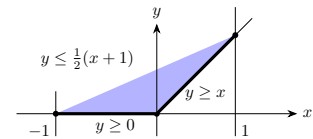

Figure 1: Triangle relaxation of a ReLU with input $x \in [-1, 1]$.

and lower bounds of $S$ (projected onto $\mathbb{R}^{d'}$) as an over-approximation of the bounds of $f(X)$. We denote by $\mathcal{C}(\boldsymbol{x}^{(1)}, \ldots, \boldsymbol{x}^{(L)})$ a set of affine constraints on the variables $\boldsymbol{x}^{(1)}, \ldots, \boldsymbol{x}^{(L)}$. Its feasible set is the intersection of the feasible set of each included affine constraint. When context is clear, we use $\mathcal{C}$ to refer to both the affine constraint set and its feasible set; for two constraint sets $\mathcal{C}_1$ and $\mathcal{C}_2$, we use $\mathcal{C}_1 \wedge \mathcal{C}_2$ to denote the combination of the constraints in $\mathcal{C}_1$ and $\mathcal{C}_2$, i.e., their feasible sets are intersected. For an affine constraint set $\mathcal{C}(\boldsymbol{x}, \boldsymbol{y})$ dependent on $\boldsymbol{x}$, we denote by $\pi_{\boldsymbol{x}}(\mathcal{C})$ the projection of the feasible set onto the $\boldsymbol{x}$-space, which can be computed by, e.g., applying the Fourier-Motzkin algorithm to remove the variables in $\mathcal{C}$ other than $\boldsymbol{x}$. We assume the domain $X$ to be a convex polytope, e.g., $L_\infty$ neighborhoods of a reference point, which is the common practice in certification. Such convex sets $S$ can be represented by a set of affine constraints $\mathcal{C}(\boldsymbol{x}, f(\boldsymbol{x}))$ as well. For example, consider the ReLU function $y = \rho(x) = \max(x, 0)$ on the domain $X = [-1, 1]$, represented by $\mathcal{C}_0 = \{x \geq -1, x \leq 1\}$. One possible convex relaxation is the Triangle relaxation (Wong & Kolter, 2018), represented by the affine constraints $\mathcal{C}_1 = \{y \geq x, y \geq 0, y \leq \frac{1}{2}(x+1)\}$. Figure 1 illustrates this, where the black thick line represents $f[X]$ and the colored area stands for $S$. In this example, $\pi_x(\mathcal{C}_0 \wedge \mathcal{C}_1) = [-1, 1]$ and $\pi_y(\mathcal{C}_0 \wedge \mathcal{C}_1) = [0, 1]$.

## 2.2 ReLU Network Analysis with Layerwise and Cross-Layer Convex Relaxations

Consider a network[1] $f = W_L \circ \rho \circ \cdots \circ \rho \circ W_1$ where $W_j$ are the affine layers for $j \in [L]$ and $\rho$ is the ReLU function. Denote the input variable by $\boldsymbol{x}$, the first layer by $\boldsymbol{v}^{(1)} := W_1(\boldsymbol{x})$, the second layer by $\boldsymbol{v}^{(2)} := \rho(\boldsymbol{v}^{(1)})$, and so on[2]. Assume the input convex polytope $X$ is defined by the affine constraint set $\mathcal{C}_0(\boldsymbol{x})$. A *layerwise convex relaxation* works as follows. Given the input convex polytope[3] $\mathcal{C}_0(\boldsymbol{x})$, apply the convex relaxation to the first layer $\boldsymbol{v}^{(1)} = W_1(\boldsymbol{x})$ to obtain a set of affine constraints $\mathcal{C}_1(\boldsymbol{x}, \boldsymbol{v}^{(1)})$. Then, based on $\pi_{\boldsymbol{v}^{(1)}}(\mathcal{C}_0 \wedge \mathcal{C}_1)$, apply it to the second layer $\boldsymbol{v}^{(2)} = \rho(\boldsymbol{v}^{(1)})$ to obtain a set of affine constraints $\mathcal{C}_2(\boldsymbol{v}^{(1)}, \boldsymbol{v}^{(2)})$. Proceeding by layer, we obtain affine constraint sets $\mathcal{C}_{j+1}(\boldsymbol{v}^{(j)}, \boldsymbol{v}^{(j+1)})$, for $j \in [2L-2]$. All the constraints pertain to a single layer and no explicit constraint across layers is allowed, e.g., $\mathcal{C}(\boldsymbol{x}, \boldsymbol{v}^{(2L-1)})$ would not appear explicitly in the above procedure. Finally, we combine all constraints to get $\mathcal{C} = \mathcal{C}_0(\boldsymbol{x}) \wedge \mathcal{C}_1(\boldsymbol{x}, \boldsymbol{v}^{(1)}) \wedge \cdots \wedge \mathcal{C}_{2L-1}(\boldsymbol{v}^{(2L-2)}, \boldsymbol{v}^{(2L-1)})$, and solve $\mathcal{C}$ to obtain the upper and lower bounds of the output variable $\boldsymbol{v}^{(2L-1)}$. These bounds are then used to certify the network.

In contrast to layerwise relaxations which consider every layer separately, *cross-layer relaxations* (Zhang et al., 2022) include constraints involving multiple consecutive layers. Concretely, let $r \in \mathbb{N}^+$, for the network $f$ above, a cross-$r$-layer relaxation processes the first $r$ layers jointly and returns a set of affine constraints $\mathcal{C}_1(\boldsymbol{x}, \boldsymbol{v}^{(1)}, \ldots, \boldsymbol{v}^{(r)})$. Proceeding again by layer, we obtain affine constraint sets $\mathcal{C}_2(\boldsymbol{v}^{(1)}, \ldots, \boldsymbol{v}^{(1+r)}), \ldots, \mathcal{C}_{2L-r}(\boldsymbol{v}^{(2L-r-1)}, \ldots, \boldsymbol{v}^{(2L-1)})$, and the intersection of all feasible sets is solved to return bounds on $\boldsymbol{v}^{(2L-1)}$. We denote by $\mathcal{P}_r$ the convex relaxation that always returns the convex hull of the function graph of every $r$ adjacent layers on an input convex polytope to the considered layers, which is, by definition, the most precise cross-$r$-layer convex relaxation, and likewise denote by $\mathcal{P}_1$ the most precise layerwise (cross-1-layer) convex relaxation. In other words, given a feasible set $S$ in the $\boldsymbol{v}^{(i)}$ space, $\mathcal{P}_r$ returns a constraint set equivalent to the convex hull of $\{(\boldsymbol{v}^{(i)}, \ldots, \boldsymbol{v}^{(i+r)}) \mid \boldsymbol{v}^{(i)} \in S\}$ for all $i$. All cross-$r$-layer relaxations cannot be made more precise than $\mathcal{P}_r$ by definition.

For a set $H$, we denote its convex hull by $\mathrm{conv}(H)$. For a compact set $X \subseteq \mathbb{R}^d$, we denote by $\min X$ the $d$-dimensional vector whose elements are the minimum value of points in $X$ on each coordinate. For example, $\min[0, 1]^2 = [0, 0]$. Given a relaxation method $\mathcal{P}$, a network $f$, and an input set $X$, we denote by $\ell(f, \mathcal{P}, X)$ the vector of lower bounds on each dimension of $f$ computed by $\mathcal{P}$ with respect to $X$; likewise we denote by $u(f, \mathcal{P}, X)$ the upper bounds. In this work, we assume linear programming is employed to solve the constraint sets generated by the convex relaxation methods, and it always returns optimal bounds based on the constraints, without indicating the existence or nonexistence of a feasible point attaining the bounds. A glossary of all notations is detailed in §B.

---

[1] Unless explicitly stated otherwise, the term *network* is understood as ReLU neural network.

[2] We consider affine transformation and ReLU as separate layers throughout the paper.

[3] We always assume the input convex polytope is non-empty.

### 2.3 SINGLE-NEURON AND MULTI-NEURON RELAXATIONS

Within the framework of layerwise convex relaxations, the optimal constraint set on an affine layer $\boldsymbol{y} = \boldsymbol{A}\boldsymbol{x} + \boldsymbol{b}$ is always $\mathcal{C}(\boldsymbol{x}, \boldsymbol{y}) = \{\boldsymbol{A}\boldsymbol{x} + \boldsymbol{b} - \boldsymbol{y} \leq \boldsymbol{0}, -\boldsymbol{A}\boldsymbol{x} - \boldsymbol{b} + \boldsymbol{y} \leq \boldsymbol{0}\}$, which translates to the equality $\boldsymbol{y} = \boldsymbol{A}\boldsymbol{x} + \boldsymbol{b}$. Such constraints introduce no loss of precision, and thus are adopted by most convex relaxation methods. Concretely, other than IBP, all convex relaxation methods considered in this paper use the exact constraints on affine layers. The core difference between relaxation methods is how they handle the ReLU function. Single-neuron relaxation methods process each ReLU neuron separately and disregard the interdependence between neurons, while multi-neuron relaxations consider a group of ReLU neurons jointly. For the vector $\boldsymbol{x}$, let $\boldsymbol{x}_i$ denote its $i$-th entry and $\boldsymbol{x}_I$ be the sub-vector of $\boldsymbol{x}$ with entries corresponding to the indices in the set $I$. For the ReLU layer $\boldsymbol{y} = \rho(\boldsymbol{x})$ with $x \in \mathbb{R}^d$, the constraint sets computed by single-neuron relaxations are of the form $\mathcal{C}(\boldsymbol{x}_i, \boldsymbol{y}_i)$ with $i \in [d]$. In contrast, multi-neuron relaxations produce constraints of the form $\mathcal{C}(\boldsymbol{x}_{I_1}, \boldsymbol{y}_{I_2})$ with $I_1, I_2 \subseteq [d]$. We only consider multi-neuron relaxations that are at least as precise as single-neuron relaxations, i.e., for every $i \in [d]$, there exist $I_1, I_2$ such that $i \in I_1 \cap I_2$.

Singh et al. (2019a) propose the first multi-neuron relaxation called $k$-ReLU. For the ReLU layer $\boldsymbol{y} = \rho(\boldsymbol{x})$, it considers at most $k$ unstable neurons jointly—we call neurons that switch their activation states within the input set as unstable, otherwise we call them stable— and returns $\mathcal{C}(\boldsymbol{x}_I, \boldsymbol{y}_I)$, with $I \subseteq [d], |I| \leq k$. However, $k$-ReLU reduces to the single-neuron Triangle relaxation in some cases (see the case study in §5.1), thus we consider a stronger multi-neuron relaxation which only restricts the number of output variables in the constraints, allowing $\mathcal{C}(\boldsymbol{x}, \boldsymbol{y})$ to be of the form $\mathcal{C}(\boldsymbol{x}, \boldsymbol{y}_I)$ with $I \subseteq [d], |I| \leq k$. Similar tricks are also used in Tjandraatmadja et al. (2020). We denote this special multi-neuron relaxation as $\mathcal{M}_k$, and assume it always computes the convex hull of $(\boldsymbol{x}, \rho(\boldsymbol{x}_I))$, while only one index set $I$ is allowed per ReLU layer. We emphasize that $\mathcal{M}_k$ is allowed to consider unstable and stable neurons together, while $k$-ReLU only considers unstable neurons and the corresponding inputs jointly, thus $\mathcal{M}_k$ is more precise even when $k$-ReLU also computes the convex hull of the considered variables. Neurons that are not considered by a multi-neuron relaxation are processed by the single-neuron Triangle relaxation. For ReLU networks of width no more than $k$, $\mathcal{M}_k$, as a layerwise relaxation, is equivalent to the most precise layerwise relaxation $\mathcal{P}_1$. We note that $\mathcal{P}_r$ is a multi-neuron relaxation by definition, for every $r \in \mathbb{N}^+$. A toy example is provided in §C to further illustrate the concepts introduced above. We refer interested readers to Baader et al. (2024) for a more detailed introduction to concrete single-neuron and multi-neuron relaxation methods.

## 3 LAYERWISE MULTI-NEURON INCOMPLETENESS

In this section, we establish the incompleteness result for layerwise multi-neuron relaxations. We consider $\mathcal{P}_1$, the most precise layerwise multi-neuron relaxation by definition, and show that it is incomplete, and the relaxation error can be arbitrarily large. This result naturally extends to all layerwise ReLU network verifiers, as they cannot be more precise than $\mathcal{P}_1$.

We start with a simple example to demonstrate the idea. Consider the input set $X = [-1, 1]^2$ and the ReLU network $f = f' \circ \rho \circ W_1$, where $f' = \rho(\boldsymbol{x}_1 - 1) + \rho(1 - \boldsymbol{x}_1) + \rho(\boldsymbol{x}_2 - 1) + \rho(1 - \boldsymbol{x}_2)$ encodes the function $f'(\boldsymbol{x}_1, \boldsymbol{x}_2) = |\boldsymbol{x}_1 - 1| + |\boldsymbol{x}_2 - 1|, \boldsymbol{x} \in \mathbb{R}^2$, and $W_1$ is the affine transformation $W_1(\boldsymbol{x}) := \begin{pmatrix} -1 & -1.5 \\ -1 & 1.5 \end{pmatrix} \boldsymbol{x} + \begin{pmatrix} -0.5 \\ -0.5 \end{pmatrix}$, for $\boldsymbol{x} \in \mathbb{R}^2$. Let $\boldsymbol{u} := \rho(W_1(\boldsymbol{x}))$. As illustrated in Figure 2, the affine layer $W_1$ and the subsequent ReLU transform the input set into the polytope union $U = \{\boldsymbol{u}_1 \geq 0, \boldsymbol{u}_2 \geq 0, \boldsymbol{u}_1 + \boldsymbol{u}_2 \leq 1\} \cup \{1 \leq \boldsymbol{u}_1 \leq 2, \boldsymbol{u}_2 = 0\} \cup \{1 \leq \boldsymbol{u}_2 \leq 2, \boldsymbol{u}_1 = 0\}$. The minimal value of $f$ on $X$ is thus $\min f(X) = \min f'(U) = 1$. However, we will show $\ell(f, \mathcal{P}_1, X) \leq 0$, hence it is impossible to obtain the exact lower bound. To see this, consider the specific point $\boldsymbol{u}^* = (1, 1)$. On one hand, since $\mathcal{P}_1$ is a sound convex relaxation, the affine constraints obtained on the layer $\rho$ and $W_1$ characterize a convex superset of $U$, thus a superset of the convex hull of $U$ which contains $\boldsymbol{u}^*$. On the other hand, since $\mathcal{P}_1$ prohibits affine constraints across nonadjacent layers, the affine constraints induced by the subsequent layers $f'$ cannot remove $\boldsymbol{u}^*$ from the feasible set (formalized later in Lemma 3.1). Hence, the returned lower bound satisfies $\ell(f, \mathcal{P}_1, X) \leq f'(\boldsymbol{u}^*) = 0$.

We observe a general phenomenon from the example above: for a ReLU network $f = f_2 \circ f_1$, where $f_1$ and $f_2$ are its subnetworks, if (1) $f_1$ maps the input set to a set $U$ whose convex hull is its strict

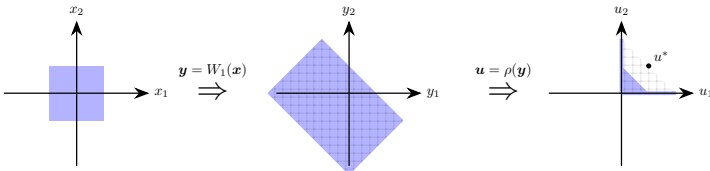

Figure 2: Blue area shows how the input box transforms under $W_1$ and ReLU; shaded area is the feasible set computed by $\mathcal{P}_1$.

superset, that is, $\text{conv}(U) \setminus U \neq \emptyset$, and (2) the subsequent network $f_2$ attains its extremal values at some point $u \in \text{conv}(U) \setminus U$, then a layerwise convex relaxation method *cannot provide* exact bounds on $f$ for the given input set. This reveals a fundamental limit of layerwise multi-neuron verifiers: there exist networks for which no verifier can provide exact bounds. In other words, all layerwise multi-neuron relaxations are incomplete, regardless of how many neurons in a single layer are jointly considered. Further, as we shall show next, the relaxation error can be unbounded. The rest of this section is devoted to formalizing and proving the ideas above.

We first establish two lemmata characterizing properties of layerwise convex relaxations. Lemma 3.1 below states that affine constraints induced by layerwise convex relaxations on some hidden layer cannot reduce the feasible set on its preceding layers.

**Lemma 3.1.** Let $L \in \mathbb{N}$ and let $X$ be a convex polytope. Consider a ReLU network $f = f_L \circ \cdots \circ f_1$. Denote the variable of the $j$-th hidden layer of $f$ by $\boldsymbol{v}^{(j)}$, for $j \in [L-1]$, and the variable of the output layer by $\boldsymbol{v}^{(L)}$. For $1 \leq i < L$, let $\mathcal{C}_1(\boldsymbol{x}, \boldsymbol{v}^{(1)}, \ldots, \boldsymbol{v}^{(i)})$ and $\mathcal{C}_2(\boldsymbol{x}, \boldsymbol{v}^{(1)}, \ldots, \boldsymbol{v}^{(L)})$ be the set of all constraints obtained by applying $\mathcal{P}_1$ to the first $i$ and $L$ layers of $f$, respectively. Then, $\pi_{\boldsymbol{v}^{(i)}}(\mathcal{C}_1(\boldsymbol{x}, \boldsymbol{v}^{(1)}, \ldots, \boldsymbol{v}^{(i)})) = \pi_{\boldsymbol{v}^{(i)}}(\mathcal{C}_2(\boldsymbol{x}, \boldsymbol{v}^{(1)}, \ldots, \boldsymbol{v}^{(L)}))$.

The proof is based on the definition of layerwise convex relaxations and is straightforward; we defer it to §E.1. Lemma 3.1 shows that the constraints induced by the deeper-than-$i$ layers do not affect the feasible set of $\boldsymbol{v}^{(i)}$. Despite the simplicity, this observation leads to Lemma 3.2, which states that the bounds computed by $\mathcal{P}_1$ cannot be better than splitting the network into two subnetworks at some hidden layer and then computing their convex hulls separately.

**Lemma 3.2.** Let $X$ be a convex polytope and consider a network $f := f_2 \circ f_1$, where $f_1$ and $f_2$ are its subnetworks. Then, $\ell(f, \mathcal{P}_1, X) \leq \min(f_2(\text{conv}(f_1(X))))$ and $u(f, \mathcal{P}_1, X) \geq \max(f_2(\text{conv}(f_1(X))))$.

The proof of Lemma 3.2 is as follows: for $f_1$, the best approximation that a convex relaxation can attain is the convex hull of the output set of $f_1$; as a consequence of Lemma 3.1, when processing $f_2$, $\mathcal{P}_1$ will take the whole set $\text{conv}(f_1(X))$ into account. Thus, the best bound that $\mathcal{P}_1$ can achieve is no better than bounding $f_2(\text{conv}(f_1(X)))$. The detailed proof of Lemma 3.2 is deferred to §E.2.

Now we are ready to show that the layerwise multi-neuron relaxation $\mathcal{P}_1$ is incomplete.

**Theorem 3.3.** Let $d \in \mathbb{N}$ and let $X$ be a convex polytope in $\mathbb{R}^d$. For every $0 < T < \infty$, there exists a ReLU network $f : \mathbb{R}^d \to \mathbb{R}$ such that $\ell(f, \mathcal{P}_1, X) \leq \min f(X) - T$, and a ReLU network $g : \mathbb{R}^d \to \mathbb{R}$ such that $u(g, \mathcal{P}_1, X) \geq \max g(X) + T$.

The proof is deferred to §E.3. Informally, we construct a network $f$ such that the convex hull of the output set of the first subnetwork is a strict superset of the output set, and the subsequent layers attain its extreme values at points outside the reachable set. The construction is similar to the example provided at the beginning of this section. Then, we can scale the weights of the output layer by a large enough constant to make the relaxation error arbitrarily large.

Theorem 3.3 is an unfortunate result for layerwise multi-neuron relaxations. It shows that every layerwise convex relaxation has a failure case where the relaxation error is arbitrarily large, though calculating them, e.g., $\mathcal{P}_1$, is already computationally expensive for large networks.

## 4 CROSS-LAYER MULTI-NEURON INCOMPLETENESS

For networks of $L$ layers, $\mathcal{P}_L$ can provide exact bounds as it computes the convex hull of the input-output function. Since $\mathcal{P}_1$ is proven incomplete in §3, the natural question is whether there exists

some $r \in \mathbb{N}^+$ for $\mathcal{P}_r$ to be complete. Instead of fixing $r$ to be a constant, we consider this question in its full generality by allowing $r$ to depend on $L$ and ask: does there exist $\alpha \in (0,1)$ such that $\mathcal{P}_{\max(1,\lfloor \alpha L \rfloor)}$ provides exact bounds for all networks with $L$ layers? Our result is rather surprising: no such $\alpha$ exists. This directly implies the incompleteness of $\mathcal{P}_r$ for all $r \in \mathbb{N}^+$. Thus, the commonly believed "single-neuron" barrier of convex relaxations is actually a misnomer, as it extends to every multi-neuron convex relaxation, and should be renamed *the universal convex barrier*.

The key insight behind our result is that for every fixed $\alpha \in (0,1)$, the cross-layer relaxation $\mathcal{P}_{\max(1,\lfloor \alpha L \rfloor)}$ shares similar limitations to $\mathcal{P}_1$ for certain networks. Formally,

**Lemma 4.1.** Let $\alpha \in (0,1), d, d', L_1, L_2 \in \mathbb{N}^+$, and $X \subseteq \mathbb{R}^d$ be a convex polytope. For every $L_1$-layer network $f_1 : \mathbb{R}^d \to \mathbb{R}^{d'}$ and $L_2$-layer network $f_2 : \mathbb{R}^{d'} \to \mathbb{R}$, there exist $L > L_1 + L_2$ and a $L$-layer network $f$ such that (i) $f(\boldsymbol{x}) = f_2 \circ f_1(\boldsymbol{x})$, for $\forall \boldsymbol{x} \in X$, and (ii) $\ell(f, \mathcal{P}_{\max(1,\lfloor \alpha L \rfloor)}, X) \le \min f_2(\mathrm{conv}(f_1(X)))$ and $u(f, \mathcal{P}_{\max(1,\lfloor \alpha L \rfloor)}, X) \ge \max f_2(\mathrm{conv}(f_1(X)))$.

Lemma 4.1 extends Lemma 3.2 to cross-layer convex relaxations. The idea behind its proof is similar to the pumping lemma: the original network $f_2 \circ f_1$ is pumped by adding dummy identity layers between $f_1$ and $f_2$. While cross-layer relaxations allow direct information exchange across layers to improve bound preciseness, the pumped dummy layers block this information exchange, thereby disabling the relaxation from providing exact bounds. The formal proof is deferred to §F.1. We note that, however, only direct information exchange between $f_1$ and $f_2$ is blocked by this construction, and the cross-layer relaxation is free to provide exact bounds for both $f_1$ and $f_2$, which is easily done by $\mathcal{P}_{\max(1,\lfloor \alpha L \rfloor)}$ when $\alpha \to 1$ for large enough $L$. This is also the key difference between layerwise and cross-layer relaxations. Nevertheless, merely blocking this information is sufficient to make the relaxation incomplete, as shown in Theorem 4.2.

**Theorem 4.2.** Let $d \in \mathbb{N}$ and let $X \subset \mathbb{R}^d$ be a convex polytope. For every $\alpha \in (0,1)$ and every constant $T > 0$, there exists a network $f : \mathbb{R}^d \to \mathbb{R}$ such that $\ell(f, \mathcal{P}_{\max(1,\lfloor \alpha L \rfloor)}, X) \le \min f(X) - T$, and a network $g : \mathbb{R}^d \to \mathbb{R}$ such that $u(g, \mathcal{P}_{\max(1,\lfloor \alpha L \rfloor)}, X) \ge \max g(X) + T$.

The proof is based on the construction when proving Theorem 3.3. Specifically, we take the construction therein and apply Lemma 4.1 to obtain a deeper network that has the same semantics. Then, since the convex hull and the exact output set of $f_1$ do not completely overlap, we use a similar argument as in the proof of Theorem 3.3 to show that the $\mathcal{P}_{\max(1,\lfloor \alpha L \rfloor)}$ relaxation is incomplete for every $\alpha \in (0,1)$. The formal proof is deferred to §F.2. This result directly extends to $\mathcal{P}_{\max(k,\lfloor \alpha L \rfloor)}$ for every constant $k \in \mathbb{N}^+$.

The implication of Theorem 4.2 is daunting: even though $\mathcal{P}_{\max(1,\lfloor \alpha L \rfloor)}$ is much more powerful than every practical convex relaxation algorithm, it is still incomplete, and the bounding error can be arbitrarily large. This shows a hard threshold in the completeness of cross-layer convex relaxation verifiers: $\mathcal{P}_{\lfloor \alpha L \rfloor}$ is complete when $\alpha = 1$ and incomplete when $\alpha < 1$.

**Beyond the ReLU activation.** While the incompleteness results we established so far are for ReLU networks, they can be naturally extended to non-polynomial activation functions such as sigmoid and tanh as follows. Recall that the extension to cross-layer incompleteness (Theorem 4.2) is based on the pumping construction of Lemma 4.1 which extends to other activations, thus it suffices to show that layerwise incompleteness extends to non-polynomial activations. The proof relies on two observations: (i) there exists a network $f$ and an input set $X$ such that $\mathrm{conv}(f(X)) \setminus f(X) \ne \emptyset$, thus there exists a nonempty open set $\Delta$ such that $\Delta \subseteq \mathrm{conv}(f(X)) \setminus f(X)$, and (ii) there exists another network $g$ such that $g(\mathrm{conv}(f(X)))$ attains its minimum only inside $\Delta$. Given a non-polynomial activation function, by the universal approximation theorem (Hornik et al., 1989), the network class is dense in the space of continuous functions, thus the first condition is easy to satisfy. The second condition can be satisfied by constructing a network that approximates a continuous function that attains its unique minimum in $\Delta$. With these two core ingredients, the rest of the proof is similar to that of ReLU networks. We defer the formal statements and proofs to §J. Further, while we focus on the absolute bounding error in the main text, the relative bounding error can also be shown to be arbitrarily large; we defer the formal statements and proofs to §I.

## 5 Making Multi-Neuron Verifiers Complete

We have shown in §3 and §4 that no multi-neuron relaxation can be complete. In this section, we study techniques to augment multi-neuron methods into complete verifiers. First, §5.1 shows that a layerwise multi-neuron relaxation, specifically $\mathcal{P}_1$, can be turned into a complete verifier by an equivalence-preserving structural transformation. While this result does not directly yield a practical algorithm, an immediate corollary is that every continuous piecewise linear function can be expressed by a ReLU network that is exactly bounded by a layerwise multi-neuron relaxation, which is unattainable by single-neuron relaxations. Second, §5.2 shows a sufficient and necessary condition for $\mathcal{P}_1$ to be complete under a convex polytope partition, and that single-neuron relaxations inherently require more partitions to be complete.

### 5.1 Completeness via Network Transformations

In this section, we consider a strong layerwise multi-neuron relaxation, namely $\mathcal{P}_1$, and show that it can be turned into a complete verifier by equivalence-preserving structural transformation of the network. Given a network $f$ to be verified, we can always construct a network $g$ equivalent to $f$ but structurally more amenable to $\mathcal{P}_1$, so as to enable exact bounds. We formally state it in Theorem 5.1.

**Theorem 5.1.** For $d, d' \in \mathbb{N}^+$, let $f : \mathbb{R}^d \to \mathbb{R}^{d'}$ be a network and let $X \subseteq \mathbb{R}^d$ be a convex polytope. There exists a network $g : \mathbb{R}^d \to \mathbb{R}^{d'}$ satisfying $g = f$ on $X$, such that $\ell(g, \mathcal{P}_1, X) = \min f(X)$ and $u(g, \mathcal{P}_1, X) = \max f(X)$.

The high-level idea is as follows: $\mathcal{P}_1$ considers constraints involving a single layer, thus we need to ensure sufficient information is passed through the hidden layers to the output layer for $\mathcal{P}_1$ to provide exact bounds. This is achieved by expanding the hidden layers of $f$ on width and making the additional neurons copy the input variable. In this way, the last layer contains sufficient information of the input and as such $\mathcal{P}_1$, which ensures the convex hull of the last layer's variables, can equivalently ensure the convex hull of $f[X]$. Detailed proof is deferred to §G.1.

Theorem 5.1 shows that $\mathcal{P}_1$ is powerful enough for complete certification if an equivalence-preserving transformation is allowed. While calculating $\mathcal{P}_1$ for the transformed network might be computationally expensive and potentially intractable, the core message from Theorem 5.1 is that the expressivity of ReLU networks is no longer limited by the relaxation. As mentioned in §1, under single-neuron relaxations, the expressivity of exactly bounded ReLU networks is limited to 1-D continuous piecewise linear functions (Baader et al., 2024): beyond 1-D, even the simple "max" function in $[0, 1]^2$ cannot be encoded by a ReLU network that is exactly bounded by single-neuron relaxations. In contrast, an immediate corollary of Theorem 5.1 is that multi-neuron preserves the full expressivity of ReLU networks as representers of general continuous piecewise linear functions:

**Corollary 5.2.** For $d \in \mathbb{N}^+$, let $f : \mathbb{R}^d \to \mathbb{R}$ be a continuous piecewise linear function, and let $X \subseteq \mathbb{R}^d$ be a convex polytope. There exists a network $g : \mathbb{R}^d \to \mathbb{R}$ satisfying $g = f$ on $X$, such that $\ell(g, \mathcal{P}_1, X) = \min f(X)$ and $u(g, \mathcal{P}_1, X) = \max f(X)$.

Corollary 5.2 shows that for every continuous piecewise linear function, there exists a ReLU network encoding it that can be exactly bounded by $\mathcal{P}_1$. In practice, a multi-neuron relaxation much weaker than $\mathcal{P}_1$ may be enough for exact bounds. To illustrate this, we examine the concrete example of the "max" function in $[0, 1]^d$ and show that $\mathcal{M}_1$ is sufficient to exactly bound a network encoding it in the following.

**Case study:** $\max(x_1, x_2, \dots, x_d)$ **can be exactly bounded by** $\mathcal{M}_1$ **in** $[0, 1]^d$**.**

First, consider the case $d = 2$. The function range is $[0, 1]$. We can represent the "max" function by the ReLU network $f = x_2 + \rho(x_1 - x_2)$, as illustrated in Figure 3. This network has width two (nodes $c$ and $d$) and one unstable neuron (node $c$). Recall that $\mathcal{M}_1$ computes the convex hull of $(\boldsymbol{x}, \rho(\boldsymbol{x}_i))$ for some $i$ for each ReLU layer.

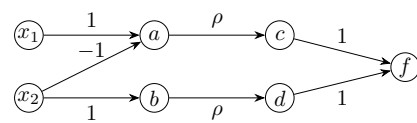

Figure 3: A network encoding $f = \max(x_1, x_2)$.

We now show that $\mathcal{M}_1$ computes the exact bounds of $f$. The input box is defined by the constraints $\{x_1 \geq 0, x_1 \leq$

$1, x_2 \geq 0, x_2 \leq 1\}$. Besides, the constraints on the affine layers are $\{a = x_1 - x_2, b = x_2, f = c + d\}$. Under these constraints, we compute bounds of the neurons of the first affine layer by linear programming, yielding $a \in [-1, 1]$ and $b \in [0, 1]$. For the stable node $d$, the constraint is then $\{d = b\}$. For the unstable node $c$, the constraint is $\{c \geq 0, c \geq a, c \leq 1 - b, c \leq a + b\}$, where the first two inequalities are based on the property of the ReLU function and the last two inequalities are based on the capability of $\mathcal{M}_1$ to compute $\text{conv}((a, b, c))$ given $\text{conv}((a, b)) = \{a \geq -b, a \leq 1 - b, b \in [0, 1], a \in [-1, 1]\}$. Note that $\text{conv}((a, b))$ is provided to $\mathcal{M}_1$ because only a single affine layer is parsed before the ReLU layer. Therefore, we have $f = c + d = c + x_2 \geq 0 + x_2 \geq 0$ and $f = c + d = c + x_2 \leq 1 - b + x_2 \leq 1$. Thus, $\mathcal{M}_1$ returns the exact upper and lower bounds. We remark that $k$-ReLU, equivalent to the Triangle relaxation in this case for every $k \geq 1$ since there is only one unstable neuron, induces on node $c$ the constraint set $\{c \geq 0, c \geq a, c \leq 0.5a + 0.5\}$. The resulting upper bound is 1.5, which is inexact, consistent to Baader et al. (2024).

Based on the 2-D case, we extend the result to $[0, 1]^d$. Indeed, we can rewrite "max" in a nested form according to $\max(x_1, x_2, \ldots, x_d) = \max(\max(x_1, x_2), \ldots, x_d)$. By the previous argument, a multi-neuron relaxation can bound $u = \max(x_1, x_2)$ exactly. Note that $u$ has no interdependency with $x_3, \ldots, x_d$, thus we can repeat the procedure above for $\max(u, x_3, \ldots, x_d)$. By induction on $d$, a multi-neuron relaxation, namely $\mathcal{M}_1$, can bound the output of a ReLU network expressing the "max" function in $[0, 1]^d$ exactly.

## 5.2 COMPLETENESS VIA CONVEX POLYTOPE PARTITIONING

In this section, we discuss how to achieve completeness for general networks (without transformation) by partitioning the input set into convex sub-polytopes.

Branch-and-bound (BaB) is currently the most effective complete verifier. It progressively divides the current problem into subproblems, solves each subproblem recursively, and combines the results to yield the bounds. With a similar strategy—we call it polytope partitioning—$\mathcal{P}_1$ can be turned into a complete verifier. The idea is to partition the input set of every layer into smaller convex polytopes so that $\mathcal{P}_1$ exactly bounds each of them. The exact bounds of the original input set can then be obtained by aggregating bounds of the smaller polytopes. An algorithm is provided in §D.

We first prove completeness, i.e., polytope partitioning enables $\mathcal{P}_1$ to calculate exact bounds.

**Proposition 5.3.** Let $L \in \mathbb{N}$ and $d_0, d_1, \ldots, d_{L+1} \in \mathbb{N}^+$. Consider an input set $X \subset \mathbb{R}^{d_0}$ and a network $f = W_{L+1} \circ \rho \circ \cdots \circ \rho \circ W_1$, where $W_j : \mathbb{R}^{d_{j-1}} \to \mathbb{R}^{d_j}$ are the associated affine transformations for $j \in [L + 1]$. Denote the subnetworks of $f$ by $f_j := W_{j+1} \circ \rho \circ \cdots \circ \rho \circ W_1$, for $j \in [L]$. Assume $H_1, \ldots, H_\nu \subseteq X$ such that $H_1, \ldots, H_\nu$ are convex polytopes, $f(X) = f(H_1) \cup \cdots \cup f(H_\nu)$, and $f_j(H_k)$ is a convex polytope for all $j \in [L]$ and $k \in [\nu]$, then

$$\min f(X) = \min_{k \in [\nu]} \ell(f, \mathcal{P}_1, H_k) \qquad \max f(X) = \max_{k \in [\nu]} u(f, \mathcal{P}_1, H_k)$$

Proposition 5.3 states that when we partition the input set into a finite collection of convex polytopes, such that each polytope remains as a convex polytope through the subsequent layers, then $\mathcal{P}_1$ can return exact bounds on the input set. The proof of Proposition 5.3 (c.f. §G.2) is based on investigating how affine and ReLU layers transform polytopes. Essentially, an affine transformation converts an input convex polytope into a convex polytope in the output space, and the ReLU function transforms a convex polytope into a union of convex polytopes. See Figure 4 for a visualization. We note that the conditions in Proposition 5.3 are not only sufficient, but also necessary: if there is a sub-polytope that is no longer a convex polytope after some layer, then the convex hull of the output set of that layer on this sub-polytope is strictly larger than the actual feasible set. From the discussion in §3, we have already known $\mathcal{P}_1$ cannot return exact bounds for general networks when this occurs.

A key question with partitioning is: what is the complexity of partitioning, that is, the number of subproblems to be solved? In particular, how does it compare with BaB when single-neuron relaxations are used for bounding? Before answering this question, we first formally define the (worst-case) partition complexity.

**Definition 5.4.** Let $\mathcal{P}$ be a complete certification method, $f$ a network, and $X$ an input set. Define the partition complexity of $\mathcal{P}$ on $f$ for $X$, denoted by #Partition($\mathcal{P}, f, X$), to be the maximum number of subproblems $\mathcal{P}$ needs to solve to compute the exact bounds of $f$ on $X$.

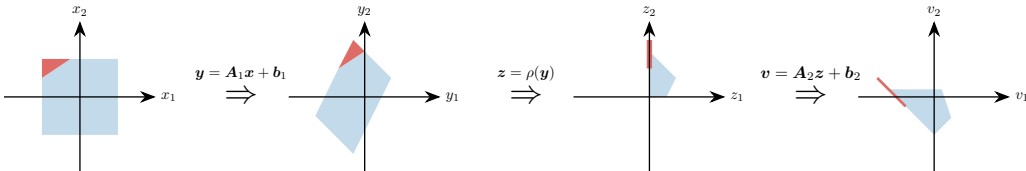

Figure 4: A partition of the input set where every part remains as a convex polytope through the layers.

**Definition 5.5.** Let $f$ be a ReLU network with $k$ ReLU neurons, and $X$ be an input set. For $x \in X$, the activation pattern of $f$ at $x$ is defined as the binary vector $\boldsymbol{a} \in \{-1, 1\}^k$ such that $\boldsymbol{a}_i = 1$ if the $i$-th ReLU neuron is activated at $x$, and $\boldsymbol{a}_i = -1$ otherwise. Denote the number of distinct activation patterns of $f$ on $X$ by $\mathcal{A}(f, X)$.

**Examples.** BaB with DEEPPOLY (Singh et al., 2019b) as the bounding method has partition complexity equal to $\mathcal{A}(f, X)$, since enumerating all possible activation patterns is both sufficient and necessary for exact bounds. BaB with IBP (Gowal et al., 2018) as the bounding method has infinite partition complexity for the network $x_1 + \rho(x_2 - x_1)$, which encodes the "max" function on $[0, 1]^2$. To see this, assume there exists a finite partition of the input set such that IBP returns exact bounds with this partition. Taking the right-upper partition, we can always find a subset of it in the form $B = [p, 1] \times [q, 1]$ for some $p, q < 1$. Then, the IBP upper bound for $x_2 - x_1$ on $B$ is $1 - p$, the IBP upper bound for $\rho(x_2 - x_1)$ is $1 - p$, and the IBP upper bound for $x_1$ is 1. Therefore, the IBP upper bound for $f$ on $B$ is at least $2 - p$, which is inexact compared to the exact upper bound 1.

In the following, we compare the partition complexity of BaB, when single-neuron relaxations and multi-neuron relaxations are used for bounding, respectively, showing that they are separated by $\mathcal{A}(f, X)$. This result holds for every single-neuron and multi-neuron relaxation in general, and does not require any assumption on the network or input set.

**Proposition 5.6.** Let $\mathcal{S}$ be some single-neuron relaxation and $\mathcal{M}$ be some multi-neuron relaxation. For every ReLU network $f$ and every input set $X$, #Partition(BaB($\mathcal{M}$), $f, X) \leq \mathcal{A}(f, X) \leq$ #Partition(BaB($\mathcal{S}$), $f, X)$.

For BaB, enumerating all possible activation patterns is necessary to obtain exact bounds even with the most precise single-neuron bounding algorithm. In contrast, Proposition 5.6 states that the activation pattern provides an upper bound on the polytope partition complexity. The proof is deferred to §G.2. Although Proposition 5.6 establishes a clear separation on partition complexity between BaB with single-neuron relaxations and multi-neuron relaxations, the upper bound can be quite conservative for powerful multi-neuron relaxations such as $\mathcal{P}_1$. We show this with a concrete example in §H, in which $\mathcal{P}_1$ with polytope partition has *exponentially smaller time complexity* than BaB with DEEPPOLY.

## 6 DISCUSSION

**Result Contexts** Our results (both negative and positive) are limited to the neural network verification setting considered in this paper. Under the given setting, the *universal convex barrier* (§3 and §4) means that for all fixed convex relaxation which may have arbitrarily large but finite resources, for all non-polynomial activation functions, all non-empty input convex polytopes and all error thresholds, there always exists a network with the activation, such that the bounding error of the relaxation on this network and input convex polytope is greater than the error threshold. This holds for both the absolute bounding error and the relative bounding error. The positive results (§5) are further limited to finite ReLU networks.

**Implications** We established a universal convex barrier, essentially ruling out the possibility of complete verifiers based solely on any convex relaxation. This implies that convex relaxations should only be applied as a subroutine in a complete verification method, such as BaB. All existing BaB methods apply single-neuron relaxations for bounding the subproblems. However, our results suggest that subproblem bounding with multi-neuron relaxations has strictly lower partition complexity. This indicates potential interest in applying efficient multi-neuron relaxations to

bound the subproblems during BaB. In addition, existing efforts on training certified models focus on single-neuron relaxations, despite the fact that none of the single-neuron relaxations can provide exact bounds for any networks encoding complex functions. In contrast, results established in §5.1 suggest that certified training with multi-neuron relaxations may be more effective, as they can provide exact bounds for every continuous piecewise linear function encoded by some networks. We leave the further investigation of practical algorithms to future work.

## 7 CONCLUSION

We conducted the first in-depth study on the expressiveness of multi-neuron convex relaxations. We extended the established single-neuron convex barrier to a *universal convex barrier* for multi-neuron relaxations, showing that they are inherently incomplete regardless of the resources allocated. On the positive side, we showed that completeness can be achieved by multi-neuron relaxations when augmented with equivalency-preserving network transformations or convex polytope partitioning, and established clear separations between multi-neuron and single-neuron relaxations in both cases. Our findings lay a solid foundation for multi-neuron relaxations and point to new directions for certified robustness.

## ACKNOWLEDGMENTS

This work has been done as part of the EU grant ELSA (European Lighthouse on Secure and Safe AI, grant agreement no. 101070617) and the SERI (Swiss State Secretariat for Education, Research and Innovation) grant SAFEAI (Certified Safe, Fair and Robust Artificial Intelligence, contract no. MB22.00088). Views and opinions expressed are however those of the authors only and do not necessarily reflect those of the European Union or European Commission. Neither the European Union nor the European Commission can be held responsible for them.

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

# A    RELATED WORK

**Neural Network Certification**    We focus on the deterministic certification of neural networks, which provides a deterministic guarantee on the robustness of a model rather than a probabilistic one (Cohen et al., 2019; Lécuyer et al., 2019; Salman et al., 2020; Carlini et al., 2023; Sun et al., 2025a;b). Existing methods for deterministic neural network certification can be categorized into complete and incomplete methods. Complete methods commonly rely on solving a mixed-integer program (Tjeng et al., 2019; Anderson et al., 2020; Tjandraatmadja et al., 2020; Tsay et al., 2021) to provide exact bounds for the output of a network. The state-of-the-art complete method (Zhang et al., 2022; Shi et al., 2025; Xu et al., 2021; Ferrari et al., 2022) is based on solving the mixed integer program with branch-and-bound (Bunel et al., 2020) on the integer variables. These methods are naturally computationally expensive and do not scale well. Incomplete methods, on the other hand, provide sound but inexact bounds, based on convex relaxations of the feasible output set of a network. Xu et al. (2020) characterizes widely-recognized single-neuron convex relaxations (Mirman et al., 2018; Wong et al., 2018; Zhang et al., 2018; Singh et al., 2019b) by their induced affine constraints, where the bounds are yielded by efficient but not necessarily optimal solvers. However, Salman et al. (2019) empirically identify a single-neuron convex barrier, preventing single-neuron relaxations from providing exact bounds for general ReLU networks, even with costly optimal solvers. To bypass this barrier, multi-neuron relaxations (Singh et al., 2018; Zhang et al., 2022; Müller et al., 2022) have been proposed and achieved higher precision empirically.

**Multi-neuron Relaxations in Practice**    To bypass the single-neuron barrier, multi-neuron relaxations (Singh et al., 2018; Zhang et al., 2022; Müller et al., 2022) have been proposed, achieving higher precision empirically. In particular, Singh et al. (2019a) and Müller et al. (2022) are looser versions of $\mathcal{P}_1$ discussed in this paper; Zhang et al. (2022) is a looser version of $\mathcal{P}_L$. Ferrari et al. (2022) combine multi-neuron relaxations with BaB and find that applying multi-neuron relaxations before BaB yields a superior overall performance. These practical applications motivate us to rigorously study the fundamental limit of multi-neuron relaxations. Furthermore, the certified training community (Müller et al., 2023; Mao et al., 2023; 2025) has already employed multi-neuron relaxations in verification, but not yet in training. This also motivates us to explore the possibility of combining multi-neuron with certified training.

**Certification with Convex Relaxations**    Existing work on the certification with convex relaxations focuses on the expressiveness of single-neuron relaxations. We distinguish three convex relaxation methods typically considered by theoretical work: Interval Bound Propagation (IBP) (Mirman et al., 2018; Gowal et al., 2018), which ignores the interdependency between neurons and use intervals $\{[a, b] \mid a, b \in \mathbb{R}\}$ for relaxation; Triangle relaxation (Wong & Kolter, 2018), which approximates the ReLU function by a triangle in the input-output space; and multi-neuron relaxations (Singh et al., 2018; Zhang et al., 2022; Müller et al., 2022) which considers a group of ReLU neurons jointly in a single affine constraint. On the positive side, Baader et al. (2020) show the universal approximation theorem for certified models, stating that for every continuous piecewise linear function $f : \mathbb{R}^n \to \mathbb{R}$ and every $\epsilon > 0$, there exists a ReLU network that approximates $f$, for which IBP provides bounds within error $\epsilon$. This result is generalized to other activations by Wang et al. (2022). However, Mirman et al. (2022) shows that there exists a continuous piecewise linear function for which IBP analysis of every finite ReLU network encoding this function provides inexact bounds. Further, Mao et al. (2024) shows that a strong regularization on the parameter signs is required for IBP to provide good bounds. Beyond IBP, Baader et al. (2024) show that even Triangle, the most precise single-neuron relaxation, cannot exactly bound any ReLU network that encodes the "max" function in $\mathbb{R}^2$, although it is provably more expressive than IBP in $\mathbb{R}$. While Baader et al. (2024) also shows that every ReLU network with a single hidden layer can be exactly bounded by multi-neuron relaxations with sufficient budget, the theoretical properties of multi-neuron relaxations in the certification of general ReLU networks remain unknown. We remark that this review is not exhaustive, especially regarding convex relaxations beyond neural network certification, and refer readers to Huchette et al. (2023) for a more comprehensive survey on MILP formulations, polyhedral geometry and expressiveness of ReLU networks.

$$\mathcal{C}_s(a,c) = \left\{ \begin{bmatrix} a-c \\ -c \\ c-\frac{1}{2}(a+1) \end{bmatrix} \le 0 \right\}$$

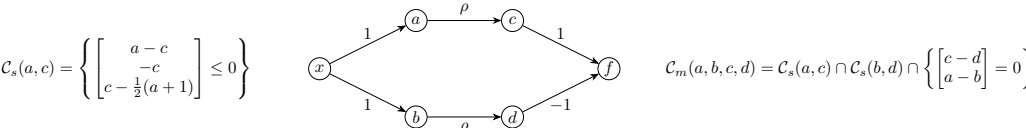

$$\mathcal{C}_m(a,b,c,d) = \mathcal{C}_s(a,c) \cap \mathcal{C}_s(b,d) \cap \left\{ \begin{bmatrix} c-d \\ a-b \end{bmatrix} = 0 \right\}$$

Figure 5: Visualization of the single-neuron and multi-neuron relaxations for a network encoding $f(x) = 0$.

## B  NOTATION

We use lowercase boldface letters to denote vectors and uppercase boldface letters to denote matrices. For the vector $\boldsymbol{x}$, $\boldsymbol{x}_i$ denotes its $i$-th entry and $\boldsymbol{x}_I$ is the subvector of $x$ with entries corresponding to the indices in the set $I$. $\boldsymbol{I}_N$ is the $N \times N$ identity matrix and $\boldsymbol{1}_N$ and $\boldsymbol{0}_N$ denotes the $N$-dimensional column vector with all entries equal to 1 and 0, respectively. $\boldsymbol{e}_i$ is the column vector with the $i$-th element taking value 1 and all other elements 0. For $\boldsymbol{1}$ and $\boldsymbol{0}$ without subscript, we understand them to be vectors of appropriate dimensions according to the context. For matrices $\boldsymbol{A}_1, \ldots, \boldsymbol{A}_n$, we designate the block-diagonal matrix with diagonal element-matrices $\boldsymbol{A}_1, \ldots, \boldsymbol{A}_n$, by $\text{diag}(\boldsymbol{A}_1, \ldots, \boldsymbol{A}_n)$.

For a set $H$, we denote the convex hull of $H$ by $\text{conv}\, H$. We represent the ReLU function as $\rho(\boldsymbol{x}) = \max(\boldsymbol{x}, \boldsymbol{0})$. For two vectors $\boldsymbol{a}, \boldsymbol{b} \in \mathbb{R}^d$, $\boldsymbol{a} \le \boldsymbol{b}$ denotes elementwise inequality.

The real set is denoted by $\mathbb{R}$, the natural numbers by $\mathbb{N}$, the positive integers by $\mathbb{N}^+$, and the $d$-dimensional real space by $\mathbb{R}^d$. For a set $S$, $|S|$ denotes the cardinality of $S$, which is the number of elements in $S$. Given $r \in \mathbb{N}^+$, $[r]$ denotes the set $\{1, 2, \ldots, r\}$. For a function $f$ and input domain $X$, we use $f(X)$ to denote its range $\{f(\boldsymbol{x}) \mid \boldsymbol{x} \in X\}$ and $f[X]$ to denote its image $\{(\boldsymbol{x}, f(\boldsymbol{x})) \mid \boldsymbol{x} \in X\}$.

We use $\mathcal{C}(\boldsymbol{x})$ to denote a set of affine constraints on $\boldsymbol{x}$, i.e., $\mathcal{C} = \{\boldsymbol{A}\boldsymbol{x} + \boldsymbol{b} \le \boldsymbol{0}\}$ for some matrix $\boldsymbol{A}$ and some vector $\boldsymbol{b}$. For two sets of constraints $\mathcal{C}_1(\boldsymbol{x}) = \{\boldsymbol{A}^{(1)}\boldsymbol{x} + \boldsymbol{b}^{(1)} \le \boldsymbol{0}\}$ and $\mathcal{C}_2(\boldsymbol{x}) = \{\boldsymbol{A}^{(2)}\boldsymbol{x} + \boldsymbol{b}^{(2)} \le \boldsymbol{0}\}$, $\mathcal{C}_1 \wedge \mathcal{C}_2 = \{\boldsymbol{A}^{(1)}\boldsymbol{x} + \boldsymbol{b}^{(1)} \le \boldsymbol{0} \wedge \boldsymbol{A}^{(2)}\boldsymbol{x} + \boldsymbol{b}^{(2)} \le \boldsymbol{0}\}$ denotes a combination of the two sets of constraints, i.e., their feasible sets are intersected.

Given a set $H = \{(\boldsymbol{x}, \boldsymbol{y}) \mid (\boldsymbol{x}, \boldsymbol{y}) \in H\}$, we denote the projection of $H$ onto the $\boldsymbol{x}$-space by $\pi_{\boldsymbol{x}}(H) = \{\boldsymbol{x} \mid \exists \boldsymbol{y} : (\boldsymbol{x}, \boldsymbol{y}) \in H\}$ and the projection onto the $\boldsymbol{y}$-space by $\pi_{\boldsymbol{y}}(H) = \{\boldsymbol{y} \mid \exists \boldsymbol{x} : (\boldsymbol{x}, \boldsymbol{y}) \in H\}$. For a feasible set $\mathcal{C}$ defined by the constraint set $\mathcal{C}(\boldsymbol{x}, \boldsymbol{y})$, $\pi_{\boldsymbol{x}}(\mathcal{C})$ is the set of values of $\boldsymbol{x}$ that satisfy the constraints in $\mathcal{C}$.

For a function $f : \mathbb{R}^{d_{\text{in}}} \to \mathbb{R}$, an input convex polytope $X \in \mathbb{R}^{d_{\text{in}}}$ and a convex relaxation $\mathcal{P}$, the lower bound of $f$ on $X$ under $\mathcal{P}$ is denoted by $\ell(f, \mathcal{P}, X)$ and the upper bound is denoted by $u(f, \mathcal{P}, X)$. Concretely, let $\mathcal{C}(\mathcal{P})$ be the constraint set induced by $\mathcal{P}$ and $v \in \mathbb{R}$ be the output variable, then $\ell(f, \mathcal{P}, X) = \min \pi_v(\mathcal{C}(\mathcal{P}))$ and $u(f, \mathcal{P}, X) = \max \pi_v(\mathcal{C}(\mathcal{P}))$.

We call neurons that switch their activation states within the input set as unstable, otherwise call it stable.

## C  EXAMPLE ILLUSTRATION

This section contains a toy example to illustrate the concepts we introduced, namely the ReLU network $\rho(x) - \rho(x)$ encoding the zero function $f(x) = 0$ with input $x \in [-1, 1]$. This network is visualized in Figure 5. The affine constraints are as follows: (i) for the input convex polytope, we have $\{x \ge -1, x \le 1\}$; (ii) for affine layers, we have $\{a = x, b = x, f = c - d\}$; (iii) for the ReLU layer, a single neuron relaxation (Triangle) will have $\mathcal{C}_s(a, c) \wedge \mathcal{C}_s(b, d)$, and a multi-neuron relaxation ($\mathcal{M}_2$) will have $\mathcal{C}_m(a, b, c, d)$. In this case, a multi-neuron relaxation successfully solves that the upper bound and lower bound of $f$ are zero, while a single-neuron relaxation solves an inexact upper bound 1 and an inexact lower bound $-1$.

# D    PSEUDO-ALGORITHM FOR POLYTOPE PARTITION

In this section, we present a pseudo-algorithm for the polytope partition in §5.2. It serves as a high-level description of the polytope partitioning algorithm. The actual implementation in practice may vary depending on the specific problem and the desired performance.

---

**Algorithm 1** Polytope Partition

---

**Input:** network $f$, input convex polytope $X$
**Output:** $u = \max_{x \in X} f(x)$ and $\ell = \min_{x \in X} f(x)$
Initialize $H \leftarrow \{(X, X)\}$
**for** each layer $f_j$ in $f$ **do**
    Initialize with a convex polytope collection $H' = \emptyset$
    **for** each pair $(H_k, S_k) \in H$ **do**
        Compute the output of $f_j$ on $S_k$, denoted by $f_j(S_k)$
        **if** $f_j(S_k)$ is a convex polytope **then**
            Add $(H_k, f_j(S_k))$ to $H'$
        **else**
            Decompose $(H_k, S_k)$ into $\nu$ convex polytopes $H_{k_1}, \ldots, H_{k_\nu}$ and the images $S_{k_1}, \ldots, S_{k_\nu}$, such that $f_j(S_{k_i})$ is a convex polytope for $i = 1, \ldots, \nu$, where $\nu$ should be as small as possible
            Add $(H_{k_i}, f_j(S_{k_i}))$ to $H'$ for $i = 1, \ldots, \nu$
        **end if**
    **end for**
    Set $H = H'$
**end for**
Initialize $\ell = +\infty$ and $u = -\infty$
**for** each convex polytope $H_k \in H$ **do**
    Update $\ell = \min(\ell, \ell(f, \mathcal{P}_1, H_k))$
    Update $u = \max(u, u(f, \mathcal{P}_1, H_k))$
**end for**
**return** $u$ and $\ell$

---

**Example.**    Running Algorithm 1 on the "max" example in §5.1, the input box $[0, 1]^d$ is always mapped to a convex polytope as it passes through the network layers. Therefore, the partition complexity is $1$.

We remark that there are two steps in the algorithm that might require high computational complexity in practice: (i) the partitioning of a set into convex polytopes, and (ii) the merging of convex polytopes. The partitioning step is necessary because the output of a ReLU network may not be a convex polytope, and we need to partition it into smaller convex polytopes to compute the bounds. The merging step is to merge redundant convex polytopes to reduce the number of subproblems. To design a practical algorithm with a low running time complexity is beyond the scope of this paper, and we leave it to the future work.

# E    DEFERRED PROOFS IN §3

## E.1    PROOF OF LEMMA 3.1

We prove Lemma 3.1, restated below for convenience.

**Lemma 3.1.**    Let $L \in \mathbb{N}$ and let $X$ be a convex polytope. Consider a ReLU network $f = f_L \circ \cdots \circ f_1$. Denote the variable of the $j$-th hidden layer of $f$ by $\boldsymbol{v}^{(j)}$, for $j \in [L-1]$, and the variable of the output layer by $\boldsymbol{v}^{(L)}$. For $1 \leq i < L$, let $\mathcal{C}_1(\boldsymbol{x}, \boldsymbol{v}^{(1)}, \ldots, \boldsymbol{v}^{(i)})$ and $\mathcal{C}_2(\boldsymbol{x}, \boldsymbol{v}^{(1)}, \ldots, \boldsymbol{v}^{(L)})$ be the set of all constraints obtained by applying $\mathcal{P}_1$ to the first $i$ and $L$ layers of $f$, respectively. Then, $\pi_{\boldsymbol{v}^{(i)}}(\mathcal{C}_1(\boldsymbol{x}, \boldsymbol{v}^{(1)}, \ldots, \boldsymbol{v}^{(i)})) = \pi_{\boldsymbol{v}^{(i)}}(\mathcal{C}_2(\boldsymbol{x}, \boldsymbol{v}^{(1)}, \ldots, \boldsymbol{v}^{(L)}))$.

*Proof.* As $\mathcal{P}$ does not consider constraints cross nonadjacent layers, $\mathcal{C}_1$ is in the form of $\mathcal{C}(\boldsymbol{x}, \boldsymbol{v}^{(1)}) \cup \mathcal{C}(\boldsymbol{v}^{(1)}, \boldsymbol{v}^{(2)}) \cup \cdots \cup \mathcal{C}(\boldsymbol{v}^{(i-1)}, \boldsymbol{v}^{(i)})$ and $\mathcal{C}_2 = \mathcal{C}_1 \cup \mathcal{C}(\boldsymbol{v}^{(i)}, \boldsymbol{v}^{(i+1)}) \cup \cdots \cup \mathcal{C}(\boldsymbol{v}^{(L-1)}, \boldsymbol{v}^{(L)})$. Let $\mathcal{C}_3 := \mathcal{C}(\boldsymbol{v}^{(i)}, \boldsymbol{v}^{(i+1)}) \cup \cdots \cup \mathcal{C}(\boldsymbol{v}^{(L-1)}, \boldsymbol{v}^{(L)})$. Note that the projection $\pi_{\boldsymbol{v}^{(i)}}(\mathcal{C}_1)$ is considered by $\mathcal{P}_1$ as the input set of the subnetwork $f_{i+1} \circ \cdots \circ f_L$ to instantiate further relaxations for deeper layers. Since $\mathcal{P}_1$ is a sound verifier, the constraints $\mathcal{C}_3$ must allow the input set, i.e.,

$$\pi_{\boldsymbol{v}^{(i)}}(\mathcal{C}_3) \supseteq \pi_{\boldsymbol{v}^{(i)}}(\mathcal{C}_1).$$

Now $\pi_{\boldsymbol{v}^{(i)}}(\mathcal{C}_2)$ is obtained by applying the Fourier-Motzkin algorithm to eliminate all the variables in $\mathcal{C}_2 = \mathcal{C}_1 \cap \mathcal{C}_3$ except $\boldsymbol{v}^{(i)}$. W.l.o.g, assume we eliminate in the following order $\boldsymbol{x}, \boldsymbol{v}^1, \ldots, \boldsymbol{v}^{(i-1)}$, $\boldsymbol{v}^{(i+1)}, \ldots, \boldsymbol{v}^{(L)}$. The constraints in $\mathcal{C}_3$ remains unchanged as we eliminate $\boldsymbol{x}, \boldsymbol{v}^1, \ldots, \boldsymbol{v}^{(i-1)}$, since they are not included in $\mathcal{C}_3$. Therefore,

$$\pi_{\boldsymbol{v}^{(i)}}(\mathcal{C}_2) = \pi_{\boldsymbol{v}^{(i)}}(\mathcal{C}_1) \cap \pi_{\boldsymbol{v}^{(i)}}(\mathcal{C}_3).$$

Hence, $\pi_{\boldsymbol{v}^{(i)}}(\mathcal{C}_2) = \pi_{\boldsymbol{v}^{(i)}}(\mathcal{C}_1)$. $\qquad\square$

### E.2 PROOF OF LEMMA 3.2

We prove Lemma 3.2, restated below for convenience.

**Lemma 3.2.** Let $X$ be a convex polytope and consider a network $f := f_2 \circ f_1$, where $f_1$ and $f_2$ are its subnetworks. Then, $\ell(f, \mathcal{P}_1, X) \leq \min(f_2(\text{conv}(f_1(X))))$ and $u(f, \mathcal{P}_1, X) \geq \max(f_2(\text{conv}(f_1(X))))$.

*Proof.* By the notation in Lemma 3.1,

$$\ell(f, \mathcal{P}_1, X) = \min_{\mu \in \pi_{\boldsymbol{v}^{(2)}}(\mathcal{C}_2(\boldsymbol{x}, \boldsymbol{v}^{(1)}, \boldsymbol{v}^{(2)}))} \mu$$

$$\leq \min_{\nu \in \pi_{\boldsymbol{v}^{(1)}}(\mathcal{C}_2(\boldsymbol{x}, \boldsymbol{v}^{(1)}, \boldsymbol{v}^{(2)}))} f_2(\nu)$$

$$= \min_{\nu \in \pi_{\boldsymbol{v}^{(1)}}(\mathcal{C}_2(\boldsymbol{x}, \boldsymbol{v}^{(1)}))} f_2(\nu),$$

where the last equality follows from Lemma 3.1. Since $\mathcal{C}_1(\boldsymbol{x}, \boldsymbol{v}^{(1)})$ is a convex polytope containing the feasible set of $\boldsymbol{v}^{(1)}$, we have $\pi_{\boldsymbol{v}^{(1)}}(\mathcal{C}_1(\boldsymbol{x}, \boldsymbol{v}^{(1)})) \supseteq \text{conv}(f_1(X))$. Therefore,

$$\ell(f, \mathcal{P}_1, X) \leq \min_{\nu \in \pi_{\boldsymbol{v}^{(1)}}(\mathcal{C}_2(\boldsymbol{x}, \boldsymbol{v}^{(1)}))} f_2(\nu)$$

$$\leq \min_{\nu \in \text{conv}(f_1(X))} f_2(\nu)$$

$$= \min(f_2(\text{conv}(f_1(X)))).$$

The proof for the upper bound is similar. $\qquad\square$

### E.3 PROOF OF THEOREM 3.3

Now we prove Theorem 3.3, restated below for convenience.

**Theorem 3.3.** Let $d \in \mathbb{N}$ and let $X$ be a convex polytope in $\mathbb{R}^d$. For every $0 < T < \infty$, there exists a ReLU network $f : \mathbb{R}^d \to \mathbb{R}$ such that $\ell(f, \mathcal{P}_1, X) \leq \min f(X) - T$, and a ReLU network $g : \mathbb{R}^d \to \mathbb{R}$ such that $u(g, \mathcal{P}_1, X) \geq \max g(X) + T$.

*Proof.* The proof is done by explicit construction of ReLU networks that satisfies the required property.

When $d = 1$, assume $X = [a, b] \subseteq \mathbb{R}$, where $a \neq b$. Let $W_0(x) = 2\frac{x-a}{b-a} - 1$, $W_1(x) = (x + 1, x)$, and $f'(\boldsymbol{x}) = 2T|\boldsymbol{x}_1 - 1| + 2T|\boldsymbol{x}_2 - 0.5| = 2T\rho(\boldsymbol{x}_1 - 1) + 2T\rho(1 - \boldsymbol{x}_1) + 2T\rho(\boldsymbol{x}_2 - 0.5) + 2T\rho(0.5 - \boldsymbol{x}_2)$, for $\boldsymbol{x} \in \mathbb{R}^2$. We construct the network as $f = f' \circ \rho \circ W_1 \circ W_0$. Since $\rho \circ W_1 \circ W_0(a) = (0, 0)$ and $\rho \circ W_1 \circ W_0(b) = (2, 1)$, $\text{conv}(\rho \circ W_1 \circ W_0([a, b])) \supseteq \{(2t, t) \mid t \in [0, 1]\}$. Thus, $\min f'(\text{conv}(\rho \circ W_1 \circ W_0([a, b]))) = 0$. Therefore, by Lemma 3.2, $\ell \leq \min f'(\text{conv}(\rho \circ W_1 \circ W_0([a, b]))) = 0$. However, the ground-truth minimum is $T$. Likewise, we can construct a ReLU network such that

applying any convex relaxation cannot provide the precise upper bound, by simply negating $f'$ to be $f'(\boldsymbol{x}) = -2T|\boldsymbol{x}_1 - 1| - 2T|\boldsymbol{x}_2 - 0.5|$.

Now assume $d \geq 2$. We assume $X$ does not degenerate, i.e., $X$ cannot be embedded in a lower-dimensional space; otherwise, we can simply project $X$ to a lower-dimensional space with a single affine layer and set $d$ to a smaller value. Now, we define the first affine layer to be the projection layer $\pi(\boldsymbol{x}) = \boldsymbol{x}_1$, which simply projects a point to its first dimension. For every non-degenerate $X$, $\pi(X)$ is a nonempty interval in $\mathbb{R}$. We then construct a ReLU network as $f = f' \circ \rho \circ W_1 \circ W_0 \circ \pi$. By the analysis above, $\ell \leq \min f'(\text{conv}(\rho \circ W_1 \circ W_0([a, b]))) - T$.

$\square$

## F    DEFERRED PROOFS IN §4

### F.1    PROOF OF LEMMA 4.1

Now we prove Lemma 4.1, restated below for convenience.

**Lemma 4.1.** Let $\alpha \in (0, 1), d, d', L_1, L_2 \in \mathbb{N}^+$, and $X \subseteq \mathbb{R}^d$ be a convex polytope. For every $L_1$-layer network $f_1 : \mathbb{R}^d \to \mathbb{R}^{d'}$ and $L_2$-layer network $f_2 : \mathbb{R}^{d'} \to \mathbb{R}$, there exist $L > L_1 + L_2$ and a $L$-layer network $f$ such that (i) $f(\boldsymbol{x}) = f_2 \circ f_1(\boldsymbol{x})$, for $\forall \boldsymbol{x} \in X$, and (ii) $\ell(f, \mathcal{P}_{\max(1, \lfloor \alpha L \rfloor)}, X) \leq \min f_2(\text{conv}(f_1(X)))$ and $u(f, \mathcal{P}_{\max(1, \lfloor \alpha L \rfloor)}, X) \geq \max f_2(\text{conv}(f_1(X)))$.

*Proof.* Intuitively, the proof is done by blocking direct information passing from $f_1$ to $f_2$ through adding dummy layers. Let $r = \max(1, \lfloor \alpha L \rfloor)$ and take

$$L = \lceil \max(\frac{1}{\alpha}, \frac{L_1 + L_2 + 1}{1 - \alpha}) \rceil \tag{1}$$

We construct the network $f$ by pumping $f_2 \circ f_1$ through adding identity layers between $f_2$ and $f_1$, thus the name pumping lemma. Concretely, let $f = f_2 \circ \underbrace{I_d \circ \cdots \circ I_d}_{(L - L_1 - L_2) \text{ times}} \circ f_1$, where $I_{d'}$ is the identify function in $\mathbb{R}^{d'}$. Take . Thus, $L - L_1 - L_2 \geq k + 1$. Denote the input variable by $\boldsymbol{v}^{(0)}$ and the variables on the $i$-th layer of $f$ by $\boldsymbol{v}^{(i)}$. By definition, $\mathcal{P}_k$ computes all constrains of the form $\mathcal{C}(\boldsymbol{v}^{(i)}, \ldots, \boldsymbol{v}^{(i+k)})$ for $i = 0, \ldots, L - k$. By the identity layer construction, we know $\boldsymbol{v}^{(L_1)} = \boldsymbol{v}^{(L_1+1)} = \ldots = \boldsymbol{v}^{(L-L_2)}$. By (1), $L - L_1 - L_2 \geq k + 1$, which means the constraints induced by $\mathcal{P}_r$ are can be reduced to constraints of the form $\mathcal{C}(\boldsymbol{v}^{(i)}, \ldots, \boldsymbol{v}^{(\min(i+r, L_1))})$, for $i = 0, \ldots, L_1$, and $\mathcal{C}(\boldsymbol{v}^{(\max(j-r, L-L_2))}, \ldots, \boldsymbol{v}^{(j)})$, for $j = L - L_2, \ldots, L$. For brevity, we slightly abuse notation and denote by $\mathcal{C}(\mathcal{P}_k)$ the union of all constraints induced by $\mathcal{P}_k$, denote by $\mathcal{C}_1$ the union of constraint sets of the form $\mathcal{C}(\boldsymbol{v}^{(i)}, \ldots, \boldsymbol{v}^{(\min(i+k, L_1))})$ for $i = 0, \ldots, L_1$, and denote by $\mathcal{C}_2$ the union of constraint sets of the form $\mathcal{C}(\boldsymbol{v}^{(\max(j-k, L-L_2))}, \ldots, \boldsymbol{v}^{(j)})$ for $j = L - L_2, \ldots, L$. Thus, $\pi_{\boldsymbol{v}^{(L-L_2)}}(\mathcal{C}(\mathcal{P}_k)) = \pi_{\boldsymbol{v}^{(L_1)}}(\mathcal{C}(\mathcal{P}_k)) = \pi_{\boldsymbol{v}^{(L_1)}}(\mathcal{C}_1)$. Since $\text{conv}(f_1(X)) \subseteq \pi_{\boldsymbol{v}^{(L_1)}}(\mathcal{C}_1)$,

$$\begin{aligned}
\ell(f, \mathcal{P}_k, X) &\leq \min f_2(\pi_{\boldsymbol{v}^{(L-L_2)}}(\mathcal{C}(\mathcal{P}_k))) \\
&= \min f_2(\pi_{\boldsymbol{v}^{(L_1)}}(\mathcal{C}_1)) \\
&\leq \min f_2(\text{conv}(f_1(X))),
\end{aligned}$$

and

$$\begin{aligned}
u(f, \mathcal{P}_k, X) &\geq \max f_2(\pi_{\boldsymbol{v}^{(L-L_2)}}(\mathcal{C}(\mathcal{P}_k))) \\
&= \max f_2(\pi_{\boldsymbol{v}^{(L_1)}}(\mathcal{C}_1)) \\
&\geq \max f_2(\text{conv}(f_1(X))).
\end{aligned}$$

$\square$

### F.2    PROOF OF THEOREM 4.2

Now we prove Theorem 4.2, restated below for convenience.

**Theorem 4.2.** Let $d \in \mathbb{N}$ and let $X \subset \mathbb{R}^d$ be a convex polytope. For every $\alpha \in (0,1)$ and every constant $T > 0$, there exists a network $f : \mathbb{R}^d \to \mathbb{R}$ such that $\ell(f, \mathcal{P}_{\max(1,\lfloor \alpha L \rfloor)}, X) \leq \min f(X) - T$, and a network $g : \mathbb{R}^d \to \mathbb{R}$ such that $u(g, \mathcal{P}_{\max(1,\lfloor \alpha L \rfloor)}, X) \geq \max g(X) + T$.

*Proof.* We reuse the construction in the proof of Theorem 3.3, augmented by Lemma 4.1. In the proof of Theorem 3.3, we constructed a feedforward network $\hat{f} := f' \circ \rho \circ W_3 \circ W_2 \circ W_1 \circ \pi$. Let $f_1 := \rho \circ W_3 \circ W_2 \circ W_1 \circ \pi$ and $f_2 := f'$. By Lemma 4.1, for some $L \in \mathbb{N}$, there exists an $L$-layer network $f$ such that $f = f_2 \circ f_1$ everywhere on $X$ and $\ell(f, \mathcal{P}_{\max(1,\lfloor \alpha L \rfloor)}, X) \leq \min f_2(\text{conv}(f_1(X))) \leq \min\{\hat{f}(x) : x \in X\} - T = \min\{f(x) : x \in X\} - T$ and $u(f, \mathcal{P}_{\max(1,\lfloor \alpha L \rfloor)}, X) \geq \max f_2(\text{conv}(f_1(X))) \geq \max\{\hat{f}(x) : x \in X\} + T = \max\{f(x) : x \in X\} + T$. $\qquad \square$

# G   DEFERRED PROOFS IN §5

## G.1   PROOF OF THEOREM 5.1 AND COROLLARY 5.2

We present a technical lemma before proving Theorem 5.1.

**Lemma G.1.** Let $H$ be a compact set in $\mathbb{R}^d$. Then, for every $i \in [d]$, $\min_{\boldsymbol{x} \in H} \boldsymbol{x}_i = \min_{\boldsymbol{v} \in \text{conv } H} \boldsymbol{v}_i$ and $\max_{\boldsymbol{x} \in H} \boldsymbol{x}_i = \max_{\boldsymbol{v} \in \text{conv } H} \boldsymbol{v}_i$.

*Proof.* We only show the equality for minimum values. The proof for maximum values is likewise.

Fix an arbitrary $i \in [d]$. Since $H \subseteq \text{conv } H$, we have

$$\min_{\boldsymbol{x} \in H} \boldsymbol{x}_i \geq \min_{\boldsymbol{v} \in \text{conv } H} \boldsymbol{v}_i. \tag{2}$$

Since the convex hull of a compact set is closed, $\exists \boldsymbol{v}^* \in \text{conv } H$ such that $\min_{\boldsymbol{v} \in \text{conv } H} \boldsymbol{v}_i = \boldsymbol{v}_i^*$. Furthermore, $\exists \boldsymbol{x}^*, \boldsymbol{y}^* \in H$ and $t \in [0,1]$, such that $\boldsymbol{v}^* = t\boldsymbol{x}^* + (1-t)\boldsymbol{y}^*$. Without loss of generality, assume $\boldsymbol{x}_i^* \leq \boldsymbol{y}_i^*$. But $\boldsymbol{x}_i^* \leq t\boldsymbol{x}_i^* + (1-t)\boldsymbol{y}_i^* = \boldsymbol{v}_i^* = \min_{\boldsymbol{v} \in \text{conv } H} \boldsymbol{v}_i$. Therefore $\min_{\boldsymbol{x} \in H} \boldsymbol{x}_i \leq \boldsymbol{x}_i^* \leq \min_{\boldsymbol{v} \in \text{conv } H} \boldsymbol{v}_i$. Combining with (2) gives $\min_{\boldsymbol{x} \in H} \boldsymbol{x}_i = \min_{\boldsymbol{v} \in \text{conv } H} \boldsymbol{v}_i$. $\quad \square$

Now we prove Theorem 5.1, restated below for convenience.

**Theorem 5.1.** For $d, d' \in \mathbb{N}^+$, let $f : \mathbb{R}^d \to \mathbb{R}^{d'}$ be a network and let $X \subseteq \mathbb{R}^d$ be a convex polytope. There exists a network $g : \mathbb{R}^d \to \mathbb{R}^{d'}$ satisfying $g = f$ on $X$, such that $\ell(g, \mathcal{P}_1, X) = \min f(X)$ and $u(g, \mathcal{P}_1, X) = \max f(X)$.

*Proof.* We construct the network $g$ based on $f$ as follows. First replicate the structure and weights of $f$ verbatim. Then add $d$ extra neurons in every hidden layer of $g$ to make copies of the input neurons. This can be achieved based on the equality $\rho(t - u) + u = t$, for $t \geq u$ and $t, u \in \mathbb{R}$. See Figure 6 for illustration. By construction, $g$ represents the same function as $f$ on $X$.

Now we prove $\mathcal{P}_1$ returns precise bounds for $g$ on $X$. Assume $g$ has $L$ layers. Denote the variables of the $i$-th hidden layer by $\boldsymbol{v}^{(j)}, j = 1, \ldots, L-1$, and the output layer by $\boldsymbol{v}^{(L)}$. By definition of $\mathcal{P}_1$, the system of constraints generated by $\mathcal{P}_1$ includes all affine constraints in the form of $\mathcal{C}(\boldsymbol{v}^{(L-1)}, \boldsymbol{v}^{(L)})$, given those passed from the $(L-1)$-th layer. Since $\boldsymbol{v}^{(L-1)}$ contains $\boldsymbol{x}$ as a part, $\mathcal{P}_1$ computes the convex hull of $g(\boldsymbol{x})$. Furthermore, by Lemma G.1, the bounds of the convex hull of the compact set $g(X)$ characterizes exact upper and lower bounds of $g(X)$. Therefore, $\mathcal{P}_1$ returns precise bounds of $g$ on $X$.

$\qquad \square$

We proceed to prove Corollary 5.2, restated below for convenience.

**Corollary 5.2.** For $d \in \mathbb{N}^+$, let $f : \mathbb{R}^d \to \mathbb{R}$ be a continuous piecewise linear function, and let $X \subseteq \mathbb{R}^d$ be a convex polytope. There exists a network $g : \mathbb{R}^d \to \mathbb{R}$ satisfying $g = f$ on $X$, such that $\ell(g, \mathcal{P}_1, X) = \min f(X)$ and $u(g, \mathcal{P}_1, X) = \max f(X)$.

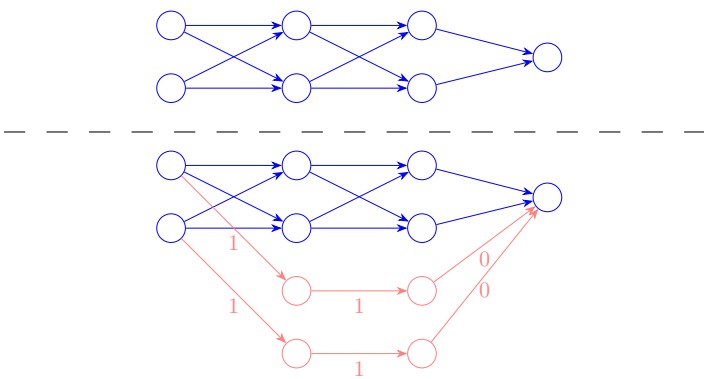

Figure 6: Top: the network $f$. Bottom: the network $g$. Labels on the edges are the associated weights.

*Proof.* For a continuous piecewise linear function $f : \mathbb{R}^d \to \mathbb{R}$, by Theorem 2.1 of Arora et al. (2018), there exists a ReLU network $g' : \mathbb{R}^d \to \mathbb{R}$ satisfying

$$f(\boldsymbol{x}) = g'(\boldsymbol{x}), \quad \boldsymbol{x} \in \mathbb{R}^d. \tag{3}$$

By Theorem 5.1, there exists another ReLU network $g : \mathbb{R}^d \to \mathbb{R}$ satisfying

$$g(\boldsymbol{x}) = g'(\boldsymbol{x}), \quad \boldsymbol{x} \in X, \tag{4}$$

and

$$\ell(g, \mathcal{P}_1, X) = \min g'(X)$$
$$u(g, \mathcal{P}_1, X) = \max g'(X).$$

Combining (3) and (4), we get

$$g(\boldsymbol{x}) = f(\boldsymbol{x}), \quad \boldsymbol{x} \in X,$$

and

$$\ell(g, \mathcal{P}_1, X) = \min f(X)$$
$$u(g, \mathcal{P}_1, X) = \max f(X).$$

$\square$

## G.2 PROOF OF PROPOSITION 5.3 AND PROPOSITION 5.6

We start with a technical lemma.

**Lemma G.2.** Let $L \in \mathbb{N}^+$. Consider a network $f = f_L \circ \cdots \circ f_1$, where $f_j$ is either an affine transformation or the ReLU function for $j \in [L]$, and an input convex polytope $X$. Denote by $f^{(j)} := f_j \circ \cdots \circ f_1$, for $j \in [L]$, the subnetworks of $f$. Assume $f^{(j)}(X)$ is a convex polytope, $\forall j \in [L]$. Then, $\ell(f, \mathcal{P}_1, X) = \min f(X)$ and $u(f, \mathcal{P}_1, X) = \max f(X)$.

*Proof.* Denote the variable of the first hidden by $\boldsymbol{v}^{(1)}$. By definition, $\mathcal{P}_1$ computes the convex hull of the function graph $(\boldsymbol{x}, \boldsymbol{v}^{(1)} = f_1(\boldsymbol{x}))$, therefore the convex hull of the feasible set of $\boldsymbol{v}^{(1)}$. Since the convex hull of a convex set is the set itself, $\mathcal{P}_1$ can precisely computes the feasible set of $\boldsymbol{v}^{(1)}$. Simply progagate by the layers and take into account the assumption that $f^{(j)}(X)$ is a convex polytope, for all $j \in [L]$, we get that $\mathcal{P}_1$ exactly bounds the network output on $X$. $\square$

We proceed to prove Proposition 5.3, restated below for convenience.

**Proposition 5.3.** Let $L \in \mathbb{N}$ and $d_0, d_1, \ldots, d_{L+1} \in \mathbb{N}^+$. Consider an input set $X \subset \mathbb{R}^{d_0}$ and a network $f = W_{L+1} \circ \rho \circ \cdots \circ \rho \circ W_1$, where $W_j : \mathbb{R}^{d_{j-1}} \to \mathbb{R}^{d_j}$ are the associated affine transformations for $j \in [L+1]$. Denote the subnetworks of $f$ by $f_j := W_{j+1} \circ \rho \circ \cdots \circ \rho \circ W_1$, for $j \in [L]$. Assume $H_1, \ldots, H_\nu \subseteq X$ such that $H_1, \ldots, H_\nu$ are convex polytopes, $f(X) = f(H_1) \cup \cdots \cup f(H_\nu)$, and $f_j(H_k)$ is a convex polytope for all $j \in [L]$ and $k \in [\nu]$, then

$$\min f(X) = \min_{k \in [\nu]} \ell(f, \mathcal{P}_1, H_k) \qquad \max f(X) = \max_{k \in [\nu]} u(f, \mathcal{P}_1, H_k)$$

*Proof.* By Lemma G.2, $\mathcal{P}_1$ returns precise bounds for $f$ on $H_k$ for all $k \in [\nu]$. Since the output set $f(X)$ is the union of $f(H_i)$ for all $k \in [\nu]$, the theorem follows. $\qquad\square$

We now prove Proposition 5.6, restated below for convenience.

**Proposition 5.6.** Let $\mathcal{S}$ be some single-neuron relaxation and $\mathcal{M}$ be some multi-neuron relaxation. For every ReLU network $f$ and every input set $X$, #Partition(BaB($\mathcal{M}$), $f, X$) $\leq \mathcal{A}(f, X) \leq$ #Partition(BaB($\mathcal{S}$), $f, X$).

*Proof.* We first prove #Partition(BaB($\mathcal{M}$), $f, X$) $\leq \mathcal{A}(f, X)$. Assume a network $f$ has $\nu :=$ $\mathcal{A}(f, X)$ distinct activation patterns on $X$. Notice that $\mathcal{M}$ always returns a constraint set that is at least as tight as DEEPPOLY, thus a same partition process as BaB(DEEPPOLY) allows BaB($\mathcal{M}$) to compute exact bounds. Recall that BaB(DEEPPOLY) has partition complexity equal to $\nu$ on $X$, therefore BaB($\mathcal{M}$) also has partition complexity at most $\nu$ on $X$.

Now we prove $\mathcal{A}(f, X) \leq$ #Partition(BaB($\mathcal{S}$), $f, X$). It suffices to show the inequality for the tightest single-neuron relaxation, i.e., the triangle relaxation, denoted by BaB($\Delta$). Given a general subproblem to bound, the only guarantee for $\Delta$ to return exact bounds is that there is no unstable neuron in the subproblem. Therefore, if BaB($\Delta$) has partition complexity equal to $K$ on $X$, then there are at most $K$ subproblems with no unstable neuron. Thus, $\mathcal{A}(f, X) \leq K$. $\qquad\square$

## H  AN EXAMPLE OF THE BENEFIT OF POLYTOPE PARTITION

For the network encoding $\max(x_1, \ldots, x_d)$ in the case study of §5.1, first note that it has $2^{d-1}$ distinct activation patterns on $[0, 1]^d$. We show that BaB requires $2^{d-1}$ branching to return precise bounds. Let $y_i = \max(x_1, \ldots, x_i)$, for $i \in [d-1]$, where $y_1 = x_1$. The $i$-th unstable neuron can then be rewritten as $\rho(y_i - x_{i+1})$, e.g., for node $c$ in Figure 3 which is the first unstable neuron, it can be rewritten as $\rho(y_1 - x_2)$. After a branching on it, this node plus $x_{i+1}$ becomes either $x_{i+1}$ when $x_{i+1} \geq y_i$, or $y_i$ when $x_{i+1} < y_i$. Therefore, this branching makes two subproblems, which are essentially the $(d-1)$-dimension "max" function. This directly implies that neither of the two subproblems can be precisely bounded by any single-neuron relaxation, thus the branching will not stop. Repeating this, BaB enumerates all $2^{d-1}$ branches, confirming the lower bound established in Proposition 5.6. In contrast, $\mathcal{P}_1$ has partition complexity 1 as shown in §5.1, leading to an exponential reduction.

Regarding the runtime, note that the number of constraints introduced by $\mathcal{P}_1$ grows linearly with $d$, while the number of branching grows exponentially with $d$ for BaB with DeepPoly. Thus, for this example, the runtime of $\mathcal{P}_1$ grows polynomially with $d$, while that of BaB with DeepPoly grows exponentially with $d$.

## I  RELATIVE BOUNDING ERROR

Theorem 3.3 and Theorem 4.2 state that the absolute bounding error by layerwise and cross-layer relaxations can be arbitrarily large. In this section, we look at the relative bounding error, namely the ratio between the length of the bounding interval and that of the exact interval. We shall show that the relative bounding error can be arbitrarily large as well. First, for $\mathcal{P}_1$, we shall prove the following statement.

**Theorem I.1.** Let $d \in \mathbb{N}$ and let $X \in \mathbb{R}^d$ be a convex polytope. For all $T > 0$, there exist a ReLU network $f : \mathbb{R}^d \to \mathbb{R}$, such that

$$\frac{u(f, \mathcal{P}_1, X) - \ell(f, \mathcal{P}_1, X)}{\max(f(X)) - \min(f(X))} > T$$

*Proof.* Without loss of generality, we prove the case when $T > 1$; otherwise, we can simply take the threshold as $\max(1, T)$ in the proof. Further, let $X = [-1, 1]$; otherwise, we can first project $X$ to one of its non-empty dimensions and scale the projected set by a single affine layer, without changing the output range of any subsequent network and the bounds computed by $\mathcal{P}_1$.

Let the ReLU network $f_1 = \rho \circ W_1$, where $W_1$ is the affine transformation $W_1(\boldsymbol{x}) := \begin{pmatrix} 1 \\ -1 \end{pmatrix} \boldsymbol{x} + \begin{pmatrix} 0 \\ 1 \end{pmatrix}$, for $\boldsymbol{x} \in \mathbb{R}^2$. The function $f_1$ maps $X$ into the set $\{\boldsymbol{x} \in \mathbb{R}^2 : \boldsymbol{x}_1 \in [0, 1], \boldsymbol{x}_2 = -\boldsymbol{x}_1 + 1\} \cup \{\boldsymbol{x} \in \mathbb{R}^2 : \boldsymbol{x}_1 = 0, 1 \le \boldsymbol{x}_2 \le 2\}$, whose convex hull is $\{\boldsymbol{x}_2 \le -2\boldsymbol{x}_1 + 2, \boldsymbol{x}_1 \ge 0, \boldsymbol{x}_2 \ge -\boldsymbol{x}_1 + 1\}$. Now consider the function $h(\boldsymbol{x}) = \boldsymbol{x}_2(\boldsymbol{x}_2 + \boldsymbol{x}_1 - 1)$, which is constantly zero on the set $f_1(X)$. We have that

$$\min(h(\operatorname{conv}(f_1(X)))) = 0, \quad \delta := \max(h(\operatorname{conv}(f_1(X)))) > 0.$$

Scaling $h$ by $2T/\delta$ gives

$$\min(\frac{2T}{\delta} h(\operatorname{conv}(f_1(X)))) = 0, \quad \max(\frac{2T}{\delta} h(\operatorname{conv}(f_1(X)))) = 2T.$$

By the universal approximation (Arora et al., 2018), there exist a ReLU network $f_2$ satisfying

$$\sup_{\operatorname{conv}(f_1(X))} |f_2 - \frac{2T}{\delta} h| \le \frac{1}{2}$$

Therefore,

$$\min(f_2 \circ f_1)(X) \ge \min(\frac{2T}{\delta} h \circ f_1)(X) - \frac{1}{2} = -\frac{1}{2},$$
$$\max(f_2 \circ f_1)(X) \le \max(\frac{2T}{\delta} h \circ f_1)(X) + \frac{1}{2} = \frac{1}{2},$$

and

$$\min f_2(\operatorname{conv}(f_1(X))) \le \min(\frac{2T}{\delta} h(\operatorname{conv}(f_1(X)))) + \frac{1}{2} = \frac{1}{2},$$
$$\max f_2(\operatorname{conv}(f_1(X))) \ge \max(\frac{2T}{\delta} h(\operatorname{conv}(f_1(X)))) - \frac{1}{2} = 2T - \frac{1}{2}.$$

Taking $f = f_2 \circ f_1$, by Lemma 3.2 we know that

$$u(f, \mathcal{P}_1, X) - \ell(f, \mathcal{P}_1, X) \ge \max f_2(\operatorname{conv}(f_1(X))) - \min f_2(\operatorname{conv}(f_1(X))) \ge 2T - 1$$

and

$$\max(f_2 \circ f_1)(X) - \min(f_2 \circ f_1)(X) \le 1.$$

Hence,

$$\frac{u(f, \mathcal{P}_1, X) - \ell(f, \mathcal{P}_1, X)}{\max(f(X)) - \min(f(X))} \ge 2T - 1 > T.$$

$\square$

We proceed to show that the relative bounding error established above for $\mathcal{P}_1$ extends to all cross-layer relaxations. Just as in §4, we do not consider specific $\mathcal{P}_r$ for some fixed $r \in \mathbb{N}$, but rather directly look at the fully general case $\mathcal{P}_{\max(1, \lfloor \alpha L \rfloor)}$ where the cross-layer is allowed to depend on the network depth $L$. Formally, we shall show

**Theorem I.2.** Let $d \in \mathbb{N}$ and let $X \in \mathbb{R}^d$ be a convex polytope. For all $T > 0$, there exist a ReLU network $f : \mathbb{R}^d \to \mathbb{R}$ of depth $L$, such that

$$\frac{u(f, \mathcal{P}_{\max(1, \lfloor \alpha L \rfloor)}, X) - \ell(f, \mathcal{P}_{\max(1, \lfloor \alpha L \rfloor)}, X)}{\max(f(X)) - \min(f(X))} > T.$$

*Proof.* Without loss of generality, we prove the case when $T > 1$; otherwise, we can simply take the threshold as $\max(1, T)$ in the proof.

We reuse the construction in the proof of Theorem I.1 and augment it by Lemma 4.1. Specifically, in the proof of Theorem I.1, we constructed a ReLU network $f = f_2 \circ f_1$ satisfying

$$\max(f(X)) - \min(f(X)) \le 1.$$

and

$$\max f_2(\operatorname{conv}(f_1(X)))(X) - \min f_2(\operatorname{conv}(f_1(X))) \ge 2T - 1$$

Now by Lemma 4.1, for some $L \in \mathbb{N}$, there exist an $L$-layer network $\hat{f}$ such that $\hat{f} = f$ everywhere on $X$ and

$$u(\hat{f}, \mathcal{P}_{\max(1, \lfloor \alpha L \rfloor)}, X) \geq \max f_2(\mathrm{conv}(f_1(X))),$$
$$\ell(\hat{f}, \mathcal{P}_{\max(1, \lfloor \alpha L \rfloor)}, X) \leq \min f_2(\mathrm{conv}(f_1(X))).$$

Therefore,

$$u(\hat{f}, \mathcal{P}_{\max(1, \lfloor \alpha L \rfloor)}, X) - \ell(\hat{f}, \mathcal{P}_{\max(1, \lfloor \alpha L \rfloor)}, X)$$
$$\geq \max f_2(\mathrm{conv}(f_1(X))) - \min f_2(\mathrm{conv}(f_1(X)))$$
$$\geq 2T - 1.$$

Hence

$$\frac{u(\hat{f}, \mathcal{P}_{\max(1, \lfloor \alpha L \rfloor)}, X) - \ell(\hat{f}, \mathcal{P}_{\max(1, \lfloor \alpha L \rfloor)}, X)}{\max(\hat{f}(X)) - \min(\hat{f}(X))}$$
$$= \frac{u(\hat{f}, \mathcal{P}_{\max(1, \lfloor \alpha L \rfloor)}, X) - \ell(\hat{f}, \mathcal{P}_{\max(1, \lfloor \alpha L \rfloor)}, X)}{\max(f(X)) - \min(f(X))}$$
$$> T.$$

$\square$

## J EXTENSION TO NON-POLYNOMIAL ACTIVATION FUNCTIONS

In this section, we extend the negative results, namely Theorem 3.3 and Theorem 4.2, established for ReLU neural networks in §3 and §4 to networks with general non-polynomial activation functions. The key insight is that by universal approximation with non-polynomial activation functions, we can always construct a network that approximates the construction for ReLU networks with arbitrary precision.

We start by introducing necessary notations. Let $H$ and $H'$ be two sets in $\mathbb{R}^d$. Then, we define the Hausdorff distance (induced by the $\ell_2$ norm) between $H$ and $H'$ as

$$D(H, H') := \max\{\sup_{\boldsymbol{x} \in H} \inf_{\boldsymbol{y} \in H'} \|\boldsymbol{x} - \boldsymbol{y}\|_2, \sup_{\boldsymbol{y} \in H'} \inf_{\boldsymbol{x} \in H} \|\boldsymbol{x} - \boldsymbol{y}\|_2\}.$$

We will use two properties of the Hausdorff distance. First, $D(H, H')$ satisfies the triangle inequality (we omit the proof since it is a standard result), i.e., for any three sets $H_1, H_2, H_3$ in $\mathbb{R}^d$, we have

$$D(H_1, H_3) \leq D(H_1, H_2) + D(H_2, H_3).$$

Second, $H \to \mathrm{conv}(H)$ is 1-Lipschitz with respect to the Hausdorff distance, stated as follows.

**Lemma J.1.** For any two sets $H_1, H_2$ in $\mathbb{R}^d$, we have

$$D(\mathrm{conv}(H_1), \mathrm{conv}(H_2)) \leq D(H_1, H_2).$$

*Proof.* We prove that $\sup_{\boldsymbol{x} \in \mathrm{conv}(H_1)} \inf_{\boldsymbol{y} \in \mathrm{conv}(H_2)} \|\boldsymbol{x} - \boldsymbol{y}\|_2 \leq D(H_1, H_2)$; the other side can be proven by symmetry.

Fix an arbitrary $\boldsymbol{x} \in \mathrm{conv}(H_1)$. By definition of convex hull, there exist $k \in \mathbb{N}^+$, $\lambda_i \geq 0$ for $i \in [k]$ with $\sum_{i=1}^k \lambda_i = 1$, and $\boldsymbol{x}_i \in H_1$ for $i \in [k]$ such that $\boldsymbol{x} = \sum_{i=1}^k \lambda_i \boldsymbol{x}_i$. By definition of Hausdorff distance, for each $i \in [k]$, there exists $\boldsymbol{y}_i \in H_2$ such that $\|\boldsymbol{x}_i - \boldsymbol{y}_i\|_2 \leq D(H_1, H_2)$. Let $\boldsymbol{y} = \sum_{i=1}^k \lambda_i \boldsymbol{y}_i$. Then, by Jensen's inequality and note that $\|\cdot\|_2$ is convex, we have

$$\|\boldsymbol{x} - \boldsymbol{y}\|_2 = \|\sum_{i=1}^k \lambda_i(\boldsymbol{x}_i - \boldsymbol{y}_i)\|_2$$
$$\leq \sum_{i=1}^k \lambda_i \|\boldsymbol{x}_i - \boldsymbol{y}_i\|_2$$
$$\leq D(H_1, H_2).$$

This implies that $\inf_{\boldsymbol{y} \in \mathrm{conv}(H_2)} \|\boldsymbol{x} - \boldsymbol{y}\|_2 \leq D(H_1, H_2)$. Since $\boldsymbol{x}$ is arbitrary, we finalize the proof. $\square$

Now we are ready to present the extended version of Theorem 3.3 for non-polynomial activation functions. We will show that for any non-polynomial activation $\sigma$, there exists a sub-network $f_1^\sigma$ and an input polytope $X$ such that $\mathrm{conv}(f_1^\sigma(X))$ is a strict superset of $f_1^\sigma(X)$. Further, for the function $f_2(x; c) := (x - c)^2$ where the point $c \in \mathrm{conv}(f_1^\sigma(X)) \setminus f_1^\sigma(X)$, there exists a network $f_2^\sigma$ approximating $f_2$ on $\mathrm{conv}(f_1^\sigma(X))$ with arbitrary precision. Combining these two results, we can construct a network $f^\sigma = f_2^\sigma \circ f_1^\sigma$ such that the bounding error by any layerwise relaxation is arbitrarily large.

**Proposition J.2.** Let $d \in \mathbb{N}$, $\sigma : \mathbb{R} \to \mathbb{R}$ be a non-polynomial activation function and $X \in \mathbb{R}^d$ be a convex polytope. Then, there exists a network $f^\sigma$, such that the $\mathrm{conv}(f^\sigma(X))$ is a strict superset of $f^\sigma(X)$.

*Proof.* Let $f_1$ be some function where $\mathrm{conv}(f_1(X)) \setminus f_1(X)$ is non-empty, e.g., the function constructed in the proof of Theorem 3.3. By universal approximation, there exists a network $f_1^\sigma$ such that

$$\sup_X \|f_1^\sigma - f_1\|_2 \le \epsilon,$$

for some $\epsilon > 0$ to be specified later. Let $H := f_1(X)$ and $H' := f_1^\sigma(X)$. This means

$$D(H, H') \le \epsilon.$$

Let $\Delta := D(\mathrm{conv}(H), H)$. Since $\mathrm{conv}(H) \setminus H$ is non-empty, we have $\Delta > 0$. By triangle inequality and Lemma J.1, we have

$$D(\mathrm{conv}(H), H) \le D(\mathrm{conv}(H), \mathrm{conv}(H')) + D(\mathrm{conv}(H'), H') + D(H', H)$$
$$\le 2D(H, H') + D(\mathrm{conv}(H'), H')$$
$$\le 2\epsilon + D(\mathrm{conv}(H'), H').$$

Thus, taking $\epsilon = \Delta/4$, we have

$$D(\mathrm{conv}(H'), H') \ge D(\mathrm{conv}(H), H) - 2\epsilon$$
$$= \frac{\Delta}{2} > 0.$$

This implies that $\mathrm{conv}(H') \setminus H'$ is non-empty, finalizing the proof. $\qquad\square$

**Theorem J.3.** Let $d \in \mathbb{N}$, $\sigma : \mathbb{R} \to \mathbb{R}$ be a non-polynomial activation function and $X \in \mathbb{R}^d$ be a convex polytope. For every constant $T > 0$, there exists a network $f^\sigma : \mathbb{R}^d \to \mathbb{R}$, such that $\ell(f^\sigma, \mathcal{P}_1, X) \le \min f^\sigma(X) - T$ and $u(f^\sigma, \mathcal{P}_1, X) \ge \max f^\sigma(X) + T$.

*Proof.* We only prove the lower bound case; the upper bound case can be proven similarly.

By Proposition J.2, there exists a network $f_1^\sigma$ and an input polytope $X$, such that $\mathrm{conv}(f_1^\sigma(X))$ is a strict superset of $f_1^\sigma(X)$. Let $c \in \mathrm{conv}(f_1^\sigma(X)) \setminus f_1^\sigma(X)$ such that $\delta := \min_{h \in f_1^\sigma(X)} \|h - c\|_2 > 0$. Let $f_2(h) := \|h - c\|_2$. Thus, we have

$$\min_{h \in f_1^\sigma(X)} f_2(h) = \delta, \qquad \min_{h \in \mathrm{conv}(f_1^\sigma(X))} f_2(h) = 0.$$

By universal approximation, there exists a network $f_2^\sigma$ such that

$$\sup_{\mathrm{conv}(f_1^\sigma(X))} |f_2^\sigma - \frac{2T}{\delta} f_2| \le \epsilon,$$

for some $\epsilon > 0$ to be specified later. Let $f^\sigma := f_2^\sigma \circ f_1^\sigma$. Then, we have

$$\min f^\sigma(X) \ge \min_{h \in f_1^\sigma(X)} \frac{2T}{\delta} f_2(h) - \epsilon = 2T - \epsilon,$$

$$\ell(f^\sigma, \mathcal{P}_1, X) \le \min_{h \in \mathrm{conv}(f_1^\sigma(X))} \frac{2T}{\delta} f_2(h) + \epsilon = \epsilon.$$

Thus, we have

$$\ell(f^\sigma, \mathcal{P}_1, X) - \min f^\sigma(X) \le -2T + 2\epsilon.$$

Let $\epsilon = T/2$, we finalize the proof. $\qquad\square$

We proceed to extend the result to cross-layer relaxations. The proof is similar to that of Theorem 4.2, where we construct dummy layers to increase the network depth without changing the network output on $X$. The only difference is that now an identity layer might not be constructed exactly, but needs to be approximated.

**Theorem J.4.** Let $d \in \mathbb{N}$ and let $X \in \mathbb{R}^d$ be a convex polytope. For every $\alpha \in (0, 1)$ and every constant $T > 0$, there exists a network $f \in \mathcal{N}^\sigma$ of depth $L$, $f : \mathbb{R}^d \to \mathbb{R}$, such that $\ell(f, \mathcal{P}_{\max(1,\alpha L)}, X) \leq \min f(X) - T$ and $u(f, \mathcal{P}_{\max(1,\alpha L)}, X) \geq \max f(X) + T$.

*Proof.* The proof directly follows that of Theorem 4.2, as long as we can construct identity layers with arbitrary precision. By universal approximation, for any $\epsilon > 0$, there exists a network $f_{\text{id}}^\sigma$ such that

$$\sup_{x \in \pi_i(X) + [-\delta, \delta]} \|f_{\text{id}}^\sigma(x) - x\| \leq \epsilon,$$

for $i \in [d]$ where $\pi_i(X)$ is the projection of $X$ onto its $i$-th dimension. By concatenating $d$ such networks in width, we constructed a network $\hat{f}_{\text{id}}^\sigma$ such that

$$\sup_{\boldsymbol{x} \in X + [-\delta, \delta]^d} \|\hat{f}_{\text{id}}^\sigma(\boldsymbol{x}) - \boldsymbol{x}\|_\infty \leq \epsilon,$$

and the every output neuron only depends on independent input neurons. Let $\epsilon_k := \frac{\epsilon}{2k^2}$ and $\delta_k := \epsilon$ for the $k$-th pseudo identity layer. Thus, by triangle inequality, the error introduced by $m$ such layers is bounded by $\sum_{k=1}^m \epsilon_k \leq \sum_{k=1}^m \frac{\epsilon}{2k^2} < \epsilon$ for any $m \in \mathbb{N}^+$. Therefore, by following the same construction in the proof of Theorem 4.2 and taking into account the $\epsilon$ approximation error introduced by the pseudo identity layers, we can finalize the proof similar to Theorem J.3.

$\square$

# K  LLM USAGE

LLMs (GPT-5) were used to polish the writing of the paper, and were not used for any other purpose.

