# OpenReview forum: "Expressiveness of Multi-Neuron Convex Relaxations in Neural Network Certification"
_ICLR.cc/2026/Conference — ICLR 2026 Poster_

### Official Review · Reviewer_ZUoG · 2025-10-28

**Soundness:** 4
**Presentation:** 3
**Contribution:** 4
**Rating:** 6
**Confidence:** 3

**Summary:**

This paper studies the theoretical expressive power of multi-neuron convex relaxations in neural network certification.
The author systematically proves for the first time that even allowing layerwise multi-neuron relaxation, such methods still cannot achieve complete certification, thereby extending the 'single-neuron convex barrier' to a universal convex barrier.
At the same time, the paper also presents positive results: through equivalency-preserving network transformations or partitioning the input domain into convex sub-polytopes, multi-neuron relaxation can achieve completeness while maintaining the full expressiveness of ReLU networks.
The author further demonstrates that its partition complexity under the branch-and-bound is lower than that of the single-neuron method.
Overall, this paper theoretically establishes the expressive boundaries of multi-neuron relaxations, clarifies its limitations and potential advantages, and provides a solid foundation for subsequent robust training and verifiable algorithms.

**Strengths:**

1. Significant Theoretical Contribution: The paper systematically analyzes for the first time the expressiveness and completeness of multi-neuron convex relaxations, introducing the concept of the "universal convex barrier," significantly extending the existing theoretical boundaries of single-neuron barriers.
2. Rigorous Analysis and Complete Proofs: The author demonstrates clear logic in formal definitions, lemmas, and theorem derivations, with strict mathematical reasoning. The core conclusions(such as the incompleteness of multi-neuron and the conditions for completeness) are all supported by rigorous proofs.
3. Both positive and negative outcomes, balanced viewpoints: While revealing the inherent limitations of multi-neuron approaches, the paper proposes two constructive schemes to achieve completeness (structural transformation and polytope partitioning), which are theoretically significant.

**Weaknesses:**

1. Insufficient discussion on feasibility: The proposed completion methods (structural transformation and polytope partitioning) are theoretically valid, but their computational complexity, scalability, and operability in large-scale networks have not been analyzed, so their practical application value remains unclear.
2. The mathematical processes and formula representations in the examples in Part Three are not clear enough, which may affect the understanding of subsequent sections.
3. The ⫋ symbol may not be clear in some fonts.

**Questions:**

1. Regarding the scope of 'universal convex barriers':Does this barrier apply to all forms of convex relaxations? Can the authors clarify its applicable boundaries and potential exceptions?
2. Measure of boundary relaxation: The text mentions that 'the relaxation error can be arbitrarily large,' and ‘relaxation error can be unbounded.’but it does not define the specific way the error is measured. Does it refer to output range error, boundary gap, or some other norm?
3. Applicability to non-ReLU activations: Although the paper claims that the conclusions can be extended to non-polynomial activations such as sigmoid and tanh, the proof is only briefly outlined. Could you provide a more detailed explanation of the key assumptions and limitations of this generalization?

---

> ### Author Response · Authors · 2025-11-23
>
> We thank Reviewer $\Rz$ for the encouraging and constructive feedback. We are happy to hear that Reviewer $\Rz$ considered our contribution significant and our analysis rigorous. In the following, we address their questions and comments in a point-to-point fashion.
>
>
> **Q1. Insufficient discussion on feasibility … computational complexity… remains unclear.**
>
> Indeed, for general networks, our analysis on the partitioning complexity cannot directly point to a runtime bound on polytope partitioning. This is also partially due to unknown runtime bound on multi-neuron relaxations. For a discussion on the runtime complexity (under specific circumstances), we refer to our reply to **Q2** of Reviewer $\Rw$.
>
>
> **Q2. The mathematical processes and formula representations in the examples in Part Three are not clear enough.**
>
> Thanks for the suggestion. We will provide cleaner notation in the final manuscript.
>
>
> **Q3. The $\subsetneqq$ symbol may not be clear in some fonts.**
>
> Thanks for the suggestion. We have replaced the usage of this notation globally.
>
> **Q4. Regarding the scope of 'universal convex barriers': Does this barrier apply to all forms of convex relaxations? Can the authors clarify its applicable boundaries and potential exceptions?**
>
> First of all, the term  "universal convex barrier" is strictly limited to the neural network verification setting considered in this paper. Convex relaxation is a broad field itself, and we certainly do not claim that the barrier identified in this paper for neural network verification also extends to other settings.
>
> Secondly, the barrier applies to the cases where convex relaxation is employed alone for network verification—no other techniques are augmented to reduce the bounding error. For example, if network transformation is added in the verification process, then the barrier easily dissolves, as shown in Sec. 5.1. Input set partitioning discussed in Sec. 5.2 is also a common technique to increase the bounding precision.
>
> Under the above setting, the "universal convex barrier" means that for *all* fixed convex relaxation, i.e., the resources are finite but can be arbitrarily large, for *all* non-polynomial activation functions, *all* non-empty input convex polytopes and *all* error thresholds, there always exists a network with the activation, such that the bounding error of the relaxation on this network and input convex polytope is greater than the error threshold. This holds for both the absolute bounding error and the relative bounding error.
>
> We thank the reviewer for this interesting question. We will add the discussion in the final manuscript.
>
>
> **Q5. Measure of boundary relaxation: The text mentions that 'the relaxation error can be arbitrarily large,' and ‘relaxation error can be unbounded.’ but it does not define the specific way the error is measured. Does it refer to output range error, boundary gap, or some other norm?**
>
> We thank the reviewer for this careful comment. At places where we use "relaxation error", we specifically mean the difference between the computed bounds on the output range and the exact bounds of the actual output range, i.e., the absolute bounding error. While this is clear in the formal theorem statement, we will define it more clearly in the text.
>
>
> **6. Applicability to non-ReLU activations: Although the paper claims that the conclusions can be extended to non-polynomial activations such as sigmoid and tanh, the proof is only briefly outlined. Could you provide a more detailed explanation of the key assumptions and limitations of this generalization?**
>
> Thanks for the suggestion. We have included a formal proof for non-polynomial activations in App. J.

---

### Official Review · Reviewer_wZ3Q · 2025-10-28

**Soundness:** 4
**Presentation:** 4
**Contribution:** 3
**Rating:** 8
**Confidence:** 3

**Summary:**

This paper provides a theoretical analysis on the properties of multi-neuron convex relaxations when used for certifying the robustness of neural networks. Specifically, the authors extend the result from previous work that single-neuron convex relaxations are incomplete and show that multi-neuron convex relaxations are also incomplete (both layerwise and cross-layer). However, the paper then discusses a way to create a transformed network from any ReLU network for which multi-neuron convex relaxations can provide complete bounds. The authors also analyze the partitioning required for complete algorithms that use multi-neuron convex relaxations and show that it requires less partitioning complexity than single-neuron relaxations. The authors conclude with a discussion of the implications of their results in practice and recommendations for future work.

**Strengths:**

- The paper is extremely well-written. The figures provide helpful visualizations for understanding the intuition behind each result.
- The authors do a great job of backing up the math and theoretical results with intuitive wording and concrete examples.
- The paper presents new theoretical results that extend past results from the single-neuron case to the more general multi-neuron case.
- The authors clearly describe the practical implications of their work as well as the potential promising avenues for future work in neural network verification algorithm design based on their theoretical results.
- The authors also discuss briefly how their results can be extended beyond ReLU networks.

**Weaknesses:**

The paper is focused largely on completeness for neural network verification. However, complete algorithms may not be necessary as long as the bounds from incomplete algorithms are tight enough.

**Questions:**

How do you think the practical implications of the theoretical results fit in with the increased complexity of creating and working with multi-neuron convex relaxations?

---

> ### Author Response · Authors · 2025-11-23
>
> We thank Reviewer $\Rw$ for the encouraging and constructive feedback. We are happy to hear that Reviewer $\Rw$ considered our paper “extremely well-written”, our results significant to the community, and the practical implications clearly discussed. In the following, we address their questions and comments in a point-to-point fashion.
>
> **Q1. The paper is focused largely on completeness for neural network verification. However, complete algorithms may not be necessary as long as the bounds from incomplete algorithms are tight enough.**
>
> Indeed, aside from (in)completeness, the bounding error caused by relaxation is of relevance.
> We, indeed, studied the error in Sec. 3 and 4. Specifically, Thm 3.3 and Thm 4.2 show that the *absolute bounding error* for every layerwise and cross-layer multi-neuron relaxations can be arbitrarily large. While not explicitly stated in the manuscript, the *relative bounding error*, i.e., the length of the bounding interval divided by the length of the exact interval, can also be arbitrarily large. We have added a discussion and the formal proof about the relative bounding error in App. I.
>
>
> **Q2. How do you think the practical implications of the theoretical results fit in with the increased complexity of creating and working with multi-neuron convex relaxations?**
>
> Given a network, the computational complexity of multi-neuron is, indeed, always higher than single-neuron, when BaB is not applied. The higher computational complexity is traded for higher bounding precision. When BaB is involved, Sec. 5.2 show that multi-neuron always require less partition than single-neuron. Thus we pointed to combining multi-neuron with the branching method—which has not been explored in practice—as a future direction. The intuition is that the lower partition complexity could balance out the computational complexity of multi-neuron, so that the overall verification complexity is controlled to some extent while achieving high precision.
>
> Theoretically speaking, because achieving precise bounds for general networks is coNP-hard, we do not expect multi-neuron to break this fundamental limit. The (potentially exponential) complexity bounds established in Sec. 5.2 are for general networks. For a concrete example, the complexity could be much lower than the theoretical upper bound, as shown in the case study at Line 361. In this specific example, the partition complexity is reduced exponentially, while the number of constraints created by multi-neuron is polynomial compared to constraints created by single-neuron. Thus, we ended up with a much better overall complexity in this specific example. We have included this discussion about overall runtime complexity in App. H.

---

> > ### Comment · Reviewer_wZ3Q · 2025-11-25
> >
> > I acknowledge the rebuttal and thank the authors for their detailed replies to my questions!

---

### Official Review · Reviewer_sc2X · 2025-10-29

**Soundness:** 3
**Presentation:** 2
**Contribution:** 1
**Rating:** 2
**Confidence:** 4

**Summary:**

The paper studies the theoretical limits of convex-relaxation–based neural network certification. Previous work established that single-neuron convex relaxations are inherently incomplete, which is the so-called single-neuron convex barrier. This paper generalizes that observation to multi-neuron and cross-layer relaxations, claiming a universal convex barrier: even the strongest finite convex relaxation cannot achieve completeness for all networks. The authors further show that completeness can be restored through either (i) equivalence-preserving network transformations or (ii) convex partitioning of the input domain, and provide some complexity analysis comparing single- and multi-neuron settings.

**Strengths:**

- The result unifies several scattered intuitions in the verification literature into a single statement.
- The theoretical arguments are self-contained and the structure of the results is clear.

**Weaknesses:**

- The central impossibility result follows almost directly from the geometric fact that convex hulls of non-convex sets are necessarily loose. A simple 2-layer, 2-neuron ReLU MLP already exhibits this property for any convex relaxation, regardless of neuron grouping or relaxation design. Thus, while the generalization to “all convex relaxations” is formally nice, it feels tautological to readers familiar with convex geometry and ReLU verification.

- The conclusion that completeness can be recovered by network reformulation or domain partitioning — is already implicit in existing verification frameworks (e.g., PRIMA, β-CROWN, Planet + BaB hybrids). The paper primarily reiterates these insights under a more general theoretical framework.

Overall, the paper's main results are mostly self-evident and insufficient for an ICLR paper.

**Questions:**

Is there any class of networks (e.g., affine-coupled, monotone, or linearly separable structures) for which completeness of convex relaxations can in fact be achieved without partitioning? It is a more interesting and non-trivial question than the result in this paper.

---

> ### Author Response · Authors · 2025-11-23
>
> We thank Reviewer $\Rs$ for their efforts in evaluating our work. Unfortunately, we noticed major misunderstanding on the challenges of the studied problem and the significance of the result. Before we address the questions, we refer the reviewer to our reply to Reviewer $\Rc$ for a very short informal account on the background and the significance of our result (~200 words at the very beginning). In particular, the discussion highlights the open questions we address in this work. In the following, we will address concrete questions raised by Reviewer $\Rs$, including detailed explanations on the reviewer’s confusion about the main results.
>
> **Q1. Does the central impossibility result follow almost directly from the geometric fact that convex hulls of non-convex sets are necessarily loose. For example, a 2-layer MLP?**
>
> Our result does not follow from the mentioned property, because computing convex hull is a *sufficient but not necessary* condition for computing exact bounds. We refer to our reply to **Q1** of Reviewer $\Rc$ for the same confusion. In short, simply proving that the relaxation does not yield the exact convex hull *does not imply* that the relaxation cannot yield exact bounds. We used a very different construction to prove that the bounds are inexact. Further, the provided 2-layer MLP example can be always exactly bounded by $\gP_2$, while a direct corollary of Thm. 4.2 shows that $\gP_2$ is also incomplete. This showcases that a naive construction similar to the provided example is insufficient for our proof.
>
>
> **Q2. Is the conclusion that completeness can be recovered by *network reformulation* already implicit in existing verification frameworks?**
>
> Precisely, (one of) our results is that every function has a representing network that can be exactly bounded by $\gP_1$. The statement in the question contains *factual errors*, because for single-neuron relaxations, literature has proven that completeness cannot be achieved by network transformation (see Line 53-56 in the manuscript). This directly implies that a detailed analysis is required for multi-neuron relaxations, as we have done.
>
> For multi-neuron relaxations, to the best of our knowledge, there is no existing result stating the completeness under equivalence-preserving network transformation.
>
> **Q3. Is the conclusion that completeness can be recovered by *domain partitioning* already implicit in existing verification frameworks?**
>
> We totally agree that domain partitioning is an existing technique to achieve complete verification. We did not claim it to be our result in the relevant section (Sec. 5.2). In particular, Sec. 5.2 (Line 391-393) explicitly mentioned that BaB is a known complete method even for single-neuron relaxations. The methods listed by the reviewer are also included in the broad related work (App. A).
>
> The discussion on input partitioning in Sec. 5.2 focused on the question: does multi-neuron offer fundamental advantage over single-neuron in terms of the required partition complexity for completeness? We formally show that this is indeed the case. One of the consequences is a future direction on using multi-neuron relaxations during BaB, which has never been discussed by existing literature.
>
> **Q4. Is there any class of networks (e.g., affine-coupled, monotone, or linearly separable structures) for which completeness of convex relaxations can in fact be achieved without partitioning?**
>
> The functional expressivity, including monotonicity and convexity, under single-neuron convex relaxation has been extensively studied in [1]. To summarize their result: (1) all 1-D convex/monotone piecewise linear functions can be expressed by a ReLU network that is precisely bounded by single-neuron relaxations, without any partitioning; (2) no single-neuron relaxation can precisely bound general convex monotone piecewise linear functions represented by any network, even in $\sR^2$, if without partitioning.
>
> We, indeed, have studied the functional expressivity of multi-neuron relaxations without partitioning, c.f. Corollary 5.2. Our result states that every piecewise linear function represented by some network can be precisely bounded by $\gP_1$. Together with the results in [1], the functional expressivity of ReLU networks under convex relaxations is almost answered fully.
>
> Regarding completeness, we have proven the sufficient and necessary condition for $\gP_1$ to be complete (Sec. 5.2), without relying on the structual assumption of the network.
>
> **Reference**
>
> [1] https://arxiv.org/abs/2311.04015

---

> > ### Comment · Reviewer_sc2X · 2025-11-26
> >
> > Thank authors for the clarification. I raised my rating accordingly.

---

### Official Review · Reviewer_cWwu · 2025-10-29

**Soundness:** 2
**Presentation:** 2
**Contribution:** 1
**Rating:** 2
**Confidence:** 4

**Summary:**

The submission analyzes the tightness a family of convex relaxations of neural networks. Convex relaxations are a fundamental tool for neural network verification, and are used to compute bounds on network (pre-)activations.
The authors focus on so-called "multi-neuron" relaxations, and in particular on $\mathcal{P}_1$, which captures the convex hull of any single network layer.
Results on its incompleteness for general networks are presented, and then followed by results on how to exploit these relaxations towards complete verification (that is, avoiding any loss of accuracy in the bounding computations).

**Strengths:**

The theoretical study of the tightness of network convex relaxations is definitely an important topic for the area.
Results-wise, I think the main contribution is Proposition 5.3, which states that $\mathcal{P}_1$ is "enough" to exactly describe any locally convex part of the network. While this is not groundbreaking, I found the result interesting, along with the discussion of the resulting partition complexity.

**Weaknesses:**

A general weakness of the paper is that it is not "operational": it is exclusively theoretical, and while a short discussion of the potential implications of the results is provided, I think these results are very far from being practically useful in the area.
A purely theoretical paper can be of course a great contribution to the literature, but I do not think the results here presented are impactful enough for that to hold.

Specifically, I think most of the presented results (except Proposition 5.3) are extremely underwhelming, as they eventually all boil down to the following statement: if the convex relaxation is not the convex hull of the entire network, the bounds will be incomplete.
Note that the fact that optimizing a linear function over the convex hull of a set $S$ will yield the same result as optimizing over $S$ is a common textbook result in convex analysis.
1) Sections 3 and 4 are devoted to showing that sequentially applying $\mathcal{P}_1$ (the convex hull of a single layer) and $\mathcal{P}_k$ (unless $k$ is the number of layers) will not yield the convex hull of the entire network. I do not quite see why it could have been the case. For instance, the Triangle relaxation is clearly the convex hull of the ReLU alone, but composing it with the preceding affine layer will not result in the convex hull of the composition.
2) Theorem 5.1 is, I believe, just a trick to basically condense the entire input-output relationships of the whole network into a single layer, for which $\mathcal{P}_1$ will then correspond to the convex hull of the entire network. In other words, the complexity of computing the network's convex hull is just hidden through the reformulation.

**Questions:**

- It feels to me that the submission is not appropriately placed within the context of the wider convex analysis literature. Analyzing the tightness of convex relaxations is for instance extremely important within Mixed-Integer Linear Programming (MILP). Given that neural network verification over piecewise-linear function is a MILP, there are relevant results from that community [1] which should be at least cited and, better, put in relation with the presented results.

- Do you see any way the lower partition complexity of multi-neuron convex relaxations could be exploited in practice over general neural networks?

[1] Strong mixed-integer programming formulations for trained neural networks, Mathematical Programming, 2020, Anderson et al.

---

> ### Author Response · Authors · 2025-11-23
>
> $\newcommand{\Rc}{\textcolor{green}{cWwu}}$
> $\newcommand{\Rs}{\textcolor{blue}{sc2X}}$
> $\newcommand{\Rw}{\textcolor{purple}{wZ3Q}}$
> $\newcommand{\Rz}{\textcolor{orange}{ZUoG}}$
> $\newcommand{\sR}{\mathbb{R}}$
> $\newcommand{\gP}{\mathcal{P}}$
>
> We thank Reviewer $\Rc$ for their efforts in evaluating our work. Unfortunately, we noticed major misunderstanding on the background and the main results of this work. Thus, we will begin the discussion with a significantly shortened account on the background of this work, followed by the informal statement of our main results and how these results tackled open questions in the community. While the short discussion is inherently informal and aims at gentler understanding, we refer to our manuscript (mostly the introduction and the concrete theorems) for the formal statements and interpretations. At the end, we provide point-to-point answers to concrete questions raised by Reviewer $\Rc$. Unless stated otherwise, all networks are ReLU networks and all functions are continuous piecewise linear functions in the discussion below.
>
> **Background of this work**: The most relevant existing results to our work are the *single-neuron convex barrier* and the *limited single-neuron expressivity*. The former established the incompleteness of single-neuron relaxations. The latter proves that *all* ReLU networks representing some function in $\sR^2$ cannot be exactly bounded by single-neuron relaxations. Further, it is known that $\gP_1$ is complete for single hidden-layer networks. Therefore, the following open questions were raised: (i) is there a complete multi-neuron relaxation for general networks, and if not, (ii) is there a multi-neuron relaxation that can bound every function represented by some network?
>
> **Main contribution of this work**: We answer these two open questions, bridging the gap in the theoretical capability of multi-neuron convex relaxations. For the first question, Sec. 3 answers that the layerwise multi-neuron relaxation (even though promising in literature) , $\gP_1$, is not complete for general networks. Sec. 4 further extends this impossibility result to all cross-layer multi-neuron relaxations and all non-polynomial activation functions, (perhaps surprisingly) establishing the *universal convex barrier*. For the second question, Thm 5.1 answers that $\gP_1$ can exactly bound every function represented by some network. We additionally analyzed some interesting properties of multi-neuron relaxations in Sec. 5.2, including the sufficient and necessary conditions for $\gP_1$ to provide exact bounds and the partition complexity separation between multi-neuron and single-neuron relaxations.

---

> > ### Author Response · Authors · 2025-11-23
> >
> > In the following, we answer questions raised by Reviewer $\Rc$ that haven’t been directly addressed by the previous discussion.
> >
> > **Q1. Does the analysis boil down to the following statement: if the convex relaxation is not the convex hull of the entire network, the bounds will be incomplete (inexact)?’’**
> >
> > First, this very statement is wrong. While getting the convex hull of the entire network guarantees exact bounds, getting exact bounds does not require computing the convex hull. For example, if all layers are monotonically increasing, the simple IBP relaxation can compute the exact bounds without knowing the convex hull of any layer.
> >
> > Second, in our analysis, we never constructed networks such that no relaxation can compute its convex hull. Instead, we first constructed a sub-network where the relaxation (if powerful enough) *could directly compute its convex hull*, but its convex hull does not match the exact reachable set. Then, we constructed the subsequent sub-network to have a single optimum at a point inside the convex hull but outside the exact reachable set, and argue that the overall network cannot be exactly bounded; we never argued that the convex hull cannot be computed, because this could not have implied inexact bounds. Further, the above analysis only works for layerwise relaxations, as in Sec. 3, and we use a different construction when analyzing cross-layer relaxations. We finally remark that for every fixed network, there always exists a cross-layer relaxation that can compute its convex hull and thus the exact bounds; our result states that for every fixed cross-layer relaxation, there always exists a network that the relaxation computes inexact bounds.
> >
> > We hope our clarification above resolves the misunderstanding of the reviewer and highlights the subtlety of the analysis.
> >
> > **Q2. Is Thm. 5.1 a trick to basically condense the entire input-output relationships of the whole network into a single layer?**
> >
> > Indeed, the proof of Thm. 5.1 is simple. However, as we discussed at the beginning, it has significant implications and answers open questions.
> >
> > **Q3: Did this work cite the literature on MILP, in particular, [1] in the review?**
> >
> > Please note that [1] is the very first paper in our reference. We further cited more literature on MILP in App. A.
> >
> > **Q4. Do you see any way the lower partition complexity of multi-neuron convex relaxations could be exploited in practice over general neural networks?**
> >
> > We thank the reviewer for raising this question. Indeed, we only bound the partition complexity, but the runtime complexity remains unknown. This is partially because no runtime bound on the multi-neuron relaxation has been known. For a discussion on the runtime complexity (under specific circumstances), we refer to our reply to **Q2** of Reviewer $\Rw$. In short, in some cases, the reduced partition complexity could be exponential while the increased runtime of each partition is polynomial, leading to exponential overall runtime savings.

---

### Meta-Review · Area_Chair_2pcL · 2026-01-15

**Summary:**

Studying tight relaxations for neural networks is an important topic in the field of neural network verification. The paper follows the research thread on convex relaxation barriers in NN verification and develops a rigorous theory of what multi-neuron convex relaxations can and cannot certify, proving a “universal convex barrier” (incompleteness with potentially unbounded error) and identifying two routes to regain completeness via equivalence-preserving network transformations or convex polytope partitioning.

A key contribution of the paper is that it cleanly formalizes multi-neuron and cross-layer relaxations and then proves strong negative results: even very powerful finite relaxations remain incomplete in general, with arbitrarily large bounding error. Another contribution is that the paper also delivers constructive, clarifying positive results, most notably that completeness can be achieved when augmenting verification with network transformations or with polytope partitioning, and that multi-neuron relaxations can yield strictly lower partition complexity than single-neuron relaxations under complete schemes such as BaB.​

The main weaknesses raised center on perceived novelty (some reviewers viewed parts of the impossibility results as “self-evident” from convex geometry) and limited operational/practical impact (the paper is primarily theoretical and does not provide a scalable algorithmic instantiation). However, after reviewing the discussions between the authors and reviewers carefully, the AC believes the low score reviewers indeed misunderstood part of the paper (and one reviewer responded and raised their rating after the authors’ clarifications). The paper’s contribution to the first rigorous, unified treatment of multi-neuron relaxation expressiveness is valuable for this field, so the AC recommends acceptance after careful consideration.

**Reviewer Concerns:**

Several substantive confusions/objections were directly addressed in the rebuttal. In particular, the authors clarified (in response to both cWwu and sc2X) why “not computing the convex hull” does not automatically imply inexact bounds, and explained that the impossibility proofs rely on constructions where the relaxation’s feasible convex set contains points that drive the objective but are not reachable, rather than merely observing that convex hulls can be loose. The rebuttal also answered literature-placement concerns by noting that the cited MILP work (Anderson et al., 2020) is already included and that additional MILP-related references appear in the appendix/related work.​ Other clarification requests appear to be addressed (at least in planned revisions). However, the broader critique that the results are too far from practice remains. The AC believes the theoretical contribution of this paper is insightful enough for acceptance, and the conclusion made in this paper may inspire future research on more operational methods.

**Reviewer Scores:**

Reviewer sc2X (score 2, reject) initially characterized the main impossibility result as tautological and the positive directions (reformulation/partitioning) as already implicit in existing frameworks, but then explicitly stated they raised their rating accordingly after the authors’ clarifications. Given that the rebuttal corrected factual misunderstandings and sharpened the novelty claim around expressiveness/completeness differences between single and multi-neuron settings, a reasonable estimate is that this reviewer would move to a borderline/positive score (e.g., 5 or 6).

Reviewer cWwu (score 2, reject) focused on contribution/impact, arguing the work is not operational and on a perceived lack of novelty. The rebuttal directly engaged these points (especially the “boils down to convex hull” critique and the literature-placement question). Still, because this reviewer did not return with an updated comment, the most plausible outcome is a modest upward adjustment at most (e.g., from 2 to 3) or no change, driven by persistent skepticism about impact and practicality. However, the AC believes the theoretical impact of this paper is strong enough for acceptance, and the concerns on practicality from this reviewer can be addressed in future work.

---

### Decision · Program_Chairs · 2026-01-26

Accept (Poster)